# Bridging ML and algorithms: comparison of hyperbolic embeddings

## Abstract

Hyperbolic embeddings are well-studied both in the machine learning and algorithm community. However, as the research proceeds independently in those two communities, comparisons and even awareness seem to be currently lacking. We compare the performance (time needed to compute embeddings) and the quality of the embeddings obtained by the popular approaches, both on real-life hierarchies and networks and simulated networks. In particular, according to our results, the algorithm by Bläsius et al (ESA 2016) is about 100 times faster than the Poincaré embeddings (NIPS 2017) and Lorentz embeddings (ICML 2018) by Nickel and Kiela, while achieving results of similar (or, in some cases, even better) quality.

## 1 Introduction

Hyperbolic geometry was originally created as a model of geometry where all the postulates of Euclid hold except the *parallel axiom*. While in Euclidean geometry, parallel lines stay at a constant distance, similar lines in hyperbolic geometry diverge exponentially. While hyperbolic geometry is sometimes used as an example of a mathematical concept that has no relation to the real world (where the parallel axiom appears to hold), the property of exponential growth has found applications in the visualization and modelling of hierarchical structures. In the machine learning community, probably the most influential paper Nickel & Kiela (2017) (*Poincaré embeddings*) shows that hyperbolic embeddings achieve impressive results compared to Euclidean and translational ones. The results have been improved even further in Nickel & Kiela (2018) (*Lorentz embeddings*) by changing the used model of hyperbolic geometry.

In the machine learning literature, this work is recognized as one of the first studies on hyperbolic embeddings. For example, according to Gu et al. (2019), "Initial works on hyperbolic embeddings include Nickel & Kiela (2017) [...]". However, it is worth noting that there is a rich history of hyperbolic embedding research that precedes this paper. Hyperbolic embeddings have been originally devised in the social network analysis community (the *Hyperbolic Random Graph* model, HRG), and the algorithmic properties of this model, including embedding techniques, have been extensively studied in the algorithm community.

Surprisingly, there appears to be limited cross-referencing between these two research communities. For example, machine learning papers we've examined rarely cite algorithmic works, and vice versa. We believe that there is valuable insight within algorithmic papers that could benefit the machine learning community. To keep the introduction short, we will highlight two papers we use for our comparisons: an $\tilde{O}(n)$ algorithm for creating hyperbolic embeddings (Bläsius et al., 2016) that predates (Nickel & Kiela, 2017), denoted as the *BFKL embedder*, and an algorithm to efficiently improve and evaluate hyperbolic embeddings using discrete methods (the *Discrete Hyperbolic Random Graph* model, DHRG, Kopczyński & Celińska-Kopczyńska (2017), published as Celińska-Kopczyńska & Kopczyński (2022)). In Nickel & Kiela (2017; 2018), the algorithm was benchmarked on the WordNet data, and benchmarked using MeanRank and MAP measures. In our experiments, BFKL turns out to be orders of magnitude faster, while achieving results of similar quality, or in some cases, better.

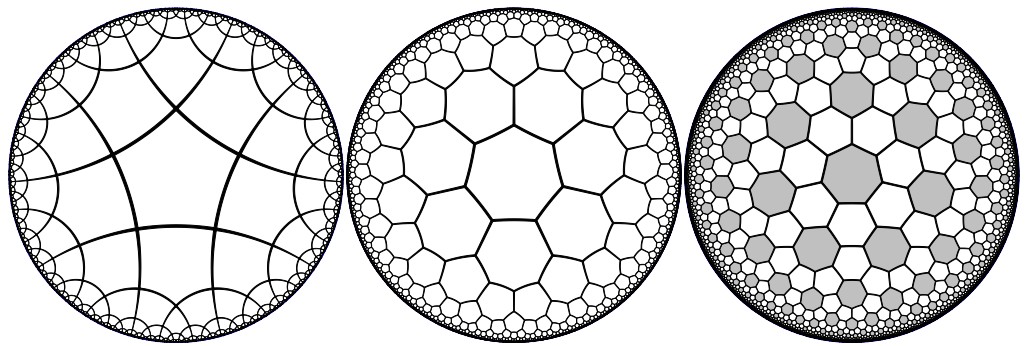

Figure 1: Tessellations of the hyperbolic plane. Bitruncated order-3 heptagonal tiling on the right.

## 2 PRELIMINARIES

### 2.1 HYPERBOLIC GEOMETRY

We start with the basics of hyperbolic geometry. For simplicity, we will focus on the hyperbolic plane $\mathbb{H}^2$, although the same ideas work in higher dimensions. See e.g. the book Cannon et al. (1997) for a more thorough formal exposition, or the game HyperRogue Kopczyński et al. (2017) to gain intuitions. Recall the Euclidean space $\mathbb{E}^n$ is $\mathbb{R}^n$ with distance $\delta_E(x,y) = \sqrt{g_+(x-y, x-y)}$, where $g_+((x_1, \ldots, x_n), (y_1, \ldots, y_n)) = \sum_{i=1}^n x_i y_i$.

In modern terms, the simplest non-Euclidean geometry is spherical geometry. A two-dimensional sphere of radius 1 is $\mathbb{S}^2 = \{x \in \mathbb{R}^3 : g_+(x,x) = 1\}$. The distance is measured in terms of great circle arcs; a point in distance $r$ in direction (angle) $\phi$ from the central point $C_0 = (0,0,1)$ has coordinates $(\sin(\phi)\sin(r), \cos(\phi)\sin(r), \cos(r))$. The spherical distance between $x$ and $y$ can be computed as $\arccos(g_+(x,y))$; this is straightforward when $y = C_0$, and also true in general, since $g_+$ is invariant under the isometries (i.e., rotations) of the sphere.

Gaussian curvature is a measure of difference of surface geometry from Euclidean geometry. A sphere of radius $R$, $R\mathbb{S}^2$, has constant Gaussian curvature $K = 1/R^2$. The hyperbolic plane is the opposite of spherical geometry, that is, it has constant negative Gaussian curvature. Hyperbolic surfaces are less ubiquitous, because they do not embed symetrically into $\mathbb{E}^3$ – that would essentially require $R$ to be imaginary. However, they appear in nature when maximizing surface area is needed (e.g., lettuce leaves), and can be embedded symetrically in the Minkowski spacetime. The hyperbolic plane $\mathbb{H}^2$ is thus $\{x \in \mathbb{R}^3 : x_3 > 0, g_-(x,x) = -1\}$, where $g_-$ is the Minkowski inner product $g_-((x_1, x_2, x_3), (y_1, y_2, y_3)) = x_1 y_1 + x_2 y_2 - x_3 y_3$ (the coordinate $x_3$ works like a time coordinate in special relativity). This is called the Minkowski hyperboloid model; many intuitions from spherical geometry work in this model, for example, a point in distance $r$ in direction (angle) $\phi$ from the central point $C_0 = (0,0,1)$ has coordinates $p(r, \phi) = (\sin(\phi)\sinh(r), \cos(\phi)\sinh(r), \cosh(r))$. The spherical distance between $x$ and $y$ can be computed as $\text{arcosh}(g_-(x,y))$.

While the formulas of the Minkowski hyperboloid model tend to be intuitively obtainable by analogy to the sphere model, this model is not applicable to visualization, since it naturally lives in Minkowski spacetime rather than the usual three-dimensional space (we use Lorentz transformations rather than Euclidean rotations for isometries involving the time coordinate). The most common method of visualization of the hyperbolic plane is the *Poincaré disk model*, first devised by Eugenio Beltrami, obtained as the stereographic projection of the Minkowski hyperboloid: $p(x,y,z) = (\frac{x}{z+1}, \frac{y}{z+1})$. This maps the (infinite) hyperbolic plane to a disk in the Euclidean plane. Figure 1 shows some tessellations of the hyperbolic plane in the Poincaré disk model. Each shape of the same shade in each of these tessellations is of the same size; the Poincaré disk model distorts distances so that the same hyperbolic distance appears smaller when closer to the boundary of the disk.

The Poincaré disk model is called a *model* (rather than *projection*) because it is often used directly, as an alternative representation of hyperbolic geometry. Many models are used; for us, the third important model is the *native polar* coordinates $(r, \phi)$. The formulas from converting from native

polar coordinates to the hyperboloid model are given above as $p(r, \phi)$. We can compute the distance formula in the native polar coordinate as follows: (let $\phi = \phi_1 - \phi_2$)

$$\delta(p(r_1, \phi_1), p(r_2, \phi_2)) = \delta(p(r_1, 0), p(r_2, \phi)) \tag{1}$$
$$= \operatorname{arcosh} g_-((\sinh(r_1), 0, \cosh(r_1)), (\sinh(r_2)\cos\phi, \sinh(r_2)\sin\phi, \cosh(r_2))) \tag{2}$$
$$= \operatorname{arcosh} (\sinh(r_1)\sinh(r_2)\cos\phi + \cosh(r_1)\cosh(r_2)) \tag{3}$$
$$= \operatorname{arcosh} (\cosh(r_1 - r_2) + (1 - \cos(\phi))\sinh(r_1)\sinh(r_2)) \tag{4}$$

The last formula has better numerical properties (Bläsius et al., 2016). The distance formula in the Poincaré disk model can be computed similarly, although converting from Poincaré to hyperboloid needs solving a quadratic equation. All models describe the same (isometric) abstract metric space, so theoretically could be equivalently used in computations, although various models differ by how robust they are to numerical precision issues (as we will see later, hyperbolic geometry exhibits exponential growth, which makes such issues very significant). All can be generalized to higher dimensions and allow interpolation between possible values of curvature $K$. In our experience, people new to computational hyperbolic geometry use Poincaré model because introductory materials often focus on it; however, they have then difficulties computing distances and isometries, while such computations are straightforward in the hyperboloid model due to the full symmetry and spherical analogies. We see the difference between Nickel & Kiela (2017) and Nickel & Kiela (2018) as an example of this. The Minkowski hyperboloid is popular as the underlying model in the visualizations of hyperbolic geometry (Phillips & Gunn, 1992; Kopczyński et al., 2017) due to simplicity and being a generalization of the *homogeneous coordinates* commonly used in computer graphics. The choice of the model may affect numerical precision (Floyd et al., 2002). As we will see later, native polar coordinates are commonly used for hyperbolic embeddings of social networks (Friedrich et al., 2023).

### 2.2 FROM VISUALIZING HIERARCHICAL DATA TO MODELLING SCALE-FREE NETWORKS

While popular expositions of hyperbolic geometry usually focus on the sum of angles of a triangle being less than 180 degrees, what is actually important to us is exponential growth. As can be easily seen from the formula for $p(r, \phi)$, a hyperbolic circle of radius $r$ has circumference $2\pi\sinh(r)$; $\sinh(r)$ grows exponentially with $r$. This exponential growth, as well as the tree-like nature of the hyperbolic space, can be seen in Figure 1, and has found application in the visualization of hierarchical data, such as trees in the hyperbolic plane (Lamping et al., 1995) and three-dimensional hyperbolic space (Munzner, 1998). Drawing a full binary tree of large depth $h$ in the Euclidean plane (say, a piece of paper), while keeping all the edges to be the same distance, is difficult, because we eventually run out of space to fit all $2^h$ leaves. The hyperbolic plane, with its exponential growth, is a perfect solution to this issue.

This leads us to another application of hyperbolic geometry, that is, the modelling of scale-free networks. Scale-free networks are commonly found in nature, technology, and as social structures. They are characterized by the *power law* distribution of degrees (the probability that a random vertex has degree $\geq d$ is proportional to $d^{-\beta}$), as well as the high *clustering coefficient* (if node $a$ is connected to $b$ and $c$, the nodes $b$ and $c$ are also likely to be connected). Despite this ubiquiteness, it is not straightforward to find a mathematical model which exhibits both these properties. One such model is the *Hyperbolic Random Graph model* (HRG) (Krioukov et al., 2010), characterized by parameters $N, R, \alpha, T$. In this model, $N$ nodes are distributed randomly in a hyperbolic disk of radius $R$. Their angular coordinates $\phi$ are distributed uniformly, while their radial coordinates $r$ are distributed according to the density function $f(r) = \alpha\sinh(\alpha r)/(\cosh(\alpha R - 1))$. Every pair of nodes $a$ and $b$ is then connected with probability $p(a, b) = (1 + \exp((\delta(a, b) - R))/2T))^{-1}$, where $\delta(a, b)$ is the hyperbolic distance between the points in $\mathbb{H}^2$ representing the two nodes. The radial coordinates corresponds to *popularity* (smaller $r$ = more popular) while the angular coordinates correspond to *similarity* (closer $\phi$ = more similar); the connections in a network are based on popularity and similarity. It can be shown that a random graph thus obtained has high clustering coefficient, and power law distribution of degrees with $\beta = 2\alpha + 1$. Hyperbolic random graphs can be generated naively in $O(n^2)$ (Aldecoa et al., 2015), in subquadratic time (von Looz et al., 2015) and in linear time (Bringmann et al., 2019). Earlier work include (Kleinberg, 2007) and (Shavitt & Tankel, 2008).

The next development is the *embedding* of real scalefree networks into the hyperbolic plane. In Boguñá et al. (2010) such an embedding of Internet was obtained, and found to be highly appropriate for *greedy routing*. In greedy routing, a node $a$ wants to find a connection to another node $b$ by

finding one of its neighbors $c$ which is the closest to $b$, then the neighbor of $c$ which is closed to $b$, and so on. Greedy routing is successful when we eventually reach $b$; the *stretch* is the ratio of the number of steps to the minimal distance between $a$ and $b$ in the network. Using greedy routing with the distances from the hyperbolic embedding achieves success rate 90%, which is significantly higher than, e.g., greedy routing based on actual geographical distances between the network nodes.

However, the embedded method used in Boguñá et al. (2010) required substantial manual intervention and did not scale to large networks (Krioukov et al., 2010). Further research focused on finding unsupervised and efficient algorithms. Usually, these algorithms are based on the maximum likelihood (MLE) method: an embedding maximizing $L$, the probability that all edges are chosen to exist or not as in the real-world network, is sought. Note that this is a difficult computational problem – even computing $L$ according to the formula requires time $O(n^2)$, which is significant for large networks. The first algorithm for embedding large networks works in time $O(n^3)$ (Papadopoulos et al., 2015b), later improved to $O(n^2)$ (Papadopoulos et al., 2015a; Wang et al., 2016).

In Bläsius et al. (2016), an quasilinear algorithm for finding hyperbolic embeddings is found. This algorithm computes the HRG parameters based on the statistics of the network. Then, it embeds the network in layers, starting from the nodes with the greatest degree, which form the center of the network. The algorithm, which we call the *BFKL embedder*, is evaluated on a number of scale-free network from the SNAP database (Leskovec & Krevl, 2014) as well as randomly generated networks generated according to the HRG model. It is shown that the greedy routing based on the BFKL embeddings again achieves good success ratio.

One embedding method is *spring embedders* (Kobourov, 2013). A spring embedder simulates forces acting on the graph: attractive forces pulling connected nodes together, and repulsive forces pushing unconnected nodes away. Spring embedders have been adapted to non-Euclidean embeddings (Kobourov, 2013), however, the straightforward adaptation to hyperbolic geometry does not produce good embeddings of large radius (Bläsius et al., 2016). The official implementation of Bläsius et al. (2016) includes a spring embedder as a method of improving the result of the quasilinear algorithm; however, the running time of this step is $\Omega(n^2)$, which is too slow for large graphs. In Celińska-Kopczyńska & Kopczyński (2022), an alternative approach is given to this problem. This approach is based on hyperbolic tilings, as shown in Figure 1 and previously used in HyperRogue Kopczyński et al. (2017). The nodes of our graph are mapped not to points of the hyperbolic plane, but rather to the tiles of such a tiling. Also, the distances are computed in a discrete way, as the number of tiles. This is called DHRG, the *discrete* HRG model. This works, because such tilings distances are a good approximation of hyperbolic distances (to a greater extent than similar approximations in Euclidean space Celińska-Kopczyńska & Kopczyński (2022)), and because the radii of HRG embeddings are large – the typical radii are on the order of $R = 30$ tiles of the bitruncated order-3 heptagonal tiling (1). One benefit of such a discrete representation is avoiding numerical precision issues. The other benefit is algorithmic: given a tile $t_1$ and a set of tiles $T$, we can compute an array $a$ such that $a[i]$ is the number of tiles in $T$ in distance $i$ from $t_1$ in time just $O(R^2)$. The time of preprocessing (add or remove a tile from $T$) is $O(R^2)$ per tile. This gives us an efficient algorithm to compute the loglikelihood of a DHRG embedding, and also to improve a DHRG embedding by local search (moving nodes to obtain a better loglikelihood).

There is extensive literature on the HRG model, for example, on its algorithmic properties. In (Bläsius et al., 2018) the impact of numerical errors on hyperbolic embeddings and greedy routing is evaluated. In Muscoloni et al. (2017); García-Pérez et al. (2019) ML algorithms are used to obtain or improve embeddings. Most research concentrates on two-dimensional embeddings. Higher-dimensional embeddings have been studied recently (Bringmann et al., 2019; Budel et al., 2023; Kovács et al., 2022; Jankowski et al., 2023).

## 2.3 Hyperbolic geometry in machine learning

In Nickel & Kiela (2017), Riemannian stochastic gradient descent (RSGD) method is applied to find hyperbolic embeddings. The algorithm is benchmarked on data that exhibits clear latent hierarchical structure (WordNet noun hierarchy) as well as on social networks (scientific collaboration communities). The quality is evaluated using MeanRank and Mean Average Precision (mAP). MeanRank is the average, over all edges $u \rightarrow v$, of $r_{u,v}$, which is the number of vertices $w$ such that there is no edges from $u$ to $w$ and $w$ is closer to $u$ than $v$ (including $u$, not including $v$, thus MeanRank

$\geq 1$). MAP is the mean of average precision scores (AP) for all vertices. The average precision score of vertex $u$ is defined as $\sum_{i=1}^{k} i/r_{u,v_i}$, where $k$ is the number of vertices $v$ such that $u \to v$, and $v_i$ is the $i$-th closest of these vertices. In case of WordNet, $u \to v$ iff $v$ is a hypernym of $u$; this is a transitive relation. In Nickel & Kiela (2018), the results are improved by using the hyperboloid model (referred to as Lorentz model) instead of Poincaré model. The results are evaluated using MeanRank, MAP, and Spearmank rank order, on multiple real-world taxonomies including the WordNet noun and verb hierarchies, the Enron email corpus, and the historical linguistics data.

In Sala et al. (2018) the effects of numerical precision is studied, more precisely, the tradeoff between the number of dimensions and the number of bits used for representing the angles. Also a combinatorial method of embedding tree-like graphs is given. In Yu & De Sa (2019), a tiling-based model (LTiling) is suggested to combat the numerical precision issues. The main idea is somewhat similar to DHRG, although while in DHRG only tiles are used, in LTiling both tiles and coordinates within the tile are used. In Gu et al. (2019) the networks are embedded not in $\mathbb{H}^n$, but in products of lower-dimensional spaces with hyperbolic, Euclidean or spherical geometry. In Chamberlain et al. (2017) hyperbolic embeddings are applied to neural networks. In Guo et al. (2022) a method for visualizing higher-dimensional hyperbolic embeddings in $\mathbb{H}^2$ is proposed.

In Nickel & Kiela (2017), the early papers on hyperbolic visualizations (Lamping et al. (1995), but not Munzner (1998)) and the HRG model are cited, although the authors and reviewers seem to not be aware of the extensive literature on hyperbolic embeddings. The Poincaré embeddings are thus compared only to Euclidean and translational embeddings. This continues in the other papers mentioned in this section. We have found citation to SNA research in Ganea et al. (2018); in Sonthalia & Gilbert (2020), Bläsius et al. (2016) is in the bibliography, but surprisingly, not referred to in text, despite the focus on speed; this paper also cites early work on hyperbolic embedding (Chepoi & Dragan, 2000), hyperbolic multi-dimensional scaling Cvetkovski & Crovella (2011), and embedding of $\delta$-hyperbolic graphs into trees (Chepoi & Dragan, 2000; Chepoi et al., 2008; Abraham et al., 2007). Comparisons between the achievements of ML and algorithm community seem to be lacking.

## 3 COMPARISON ON REAL-WORLD TAXONOMIES AND SCALE-FREE NETWORKS

Our experiment uses the following setup.

- Take a graph. Apply the Lorentz embedding LE (since it is better than Poincaré embedding PE). Since the BFKL embedder uses two-dimensional representations, we use two-dimensional embedding.
- Run the BFKL embedder on the same data.
- Compare the results of Lorentz and BFKL embedding, according to the MeanRank and MAP measures.
- Apply the DHRG embedding improvement (using the bitruncated order-3 heptagonal tiling, Figure 1) to both, and again evaluate according to our measures. (We use the discrete versions; contrary to Celińska-Kopczyńska & Kopczyński (2022) we do no dediscretization.)

We use the official implementations and hyperparameters (see Appendix A). The hyperparameters for SGD Euclidean embeddings are not given in the current official repository; we use the same parameters as for Poincaré (learning rate 1). The details are given in the supplementary material.

An implementation of MeanRank and MAP is available with Nickel & Kiela (2018). However, on most graphs, this implementation fails to evaluate the BFKL embedding due to a numerical precision error. Lorentz embeddings use a disk of radius of up to 30 bitruncated tiles (based on the constant used in the official implementation), while the BFKL embedding computes the appropriate radius for this network as 34.43 absolute units, which is 44 tiles. This larger radius leads to numerical precision errors. Therefore, we use our own implementation of these evaluations, based on the more precise distance function (4). Still, the computation is somewhat slow: for each of $O(n)$ nodes, $O(n)$ distances from the other nodes need to be computed and sorted. Therefore, we also apply the discretization from DHRG. As already mentioned, discretization allows us to compute, for every node $t$, an array $a$ such that $a[i]$ is the number of tiles in $T$ in distance $i$ from $t$, in time $O(R^2)$. If $t$ has $e_t$ edges, we can compute a similar array $b[i]$ restricted to connected tiles in time $O(e_t R)$. Note that the formulas for MeanRank and mAP given in 2.3 are for the case of continuous distances, and

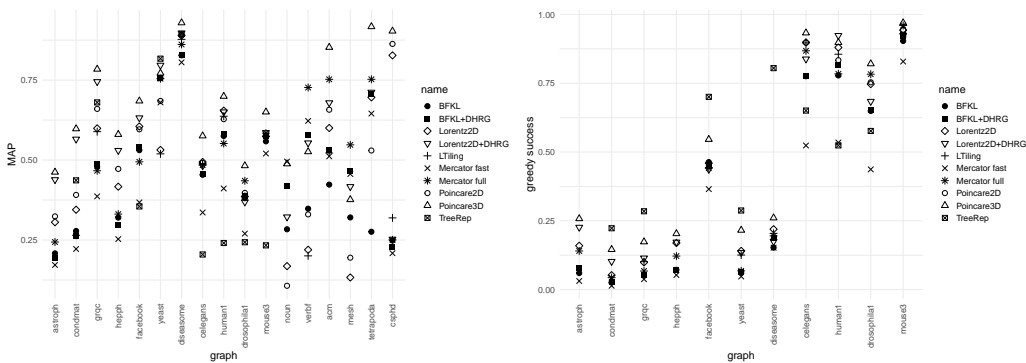

Figure 2: Quality assessment of embedders on real-world networks.

need to be adjusted for discrete values obtained from the DHRG model. In the case of MeanRank, a non-edge with distance tie contributes 0.5 to $r_{u,v}$, and in the case of mAP, if there are $b[d]$ edges and $a[d]$ total nodes in distance $d$, we assume $k$-th of these edges to be ranked after $a[d](k-0.5)/b[d]$ nodes. We can compute such MeanRank and MAP knowing $a[i]$ and $b[i]$ for every node in total time $O(nR^2 + mR)$, where $m$ is the number of edges. We can expect the scores thus obtained to be less extreme than their continuous equivalents due to lower precision. Because of this, we will refer to the discrete counterparts of mAP and MeanRank as dmAP and dMR.

We start with the WordNet hypernymy structure that PE/LE have been benchmarked on. We get mAP of 0.284 using BFKL which is significantly better than the result of Poincaré 2D of 0.118, but not the result of Lorentz 2D of 0.305, according to Nickel & Kiela (2018). However, the results obtained by us are different: 0.107 for Poincaré 2D and 0.168 for Lorentz 2D. Furthermore, while the PE/LE papers mention the good performance of their embedding methods, on our machine, BFKL is almost 100 times faster than Lorentz embedding, which is especially impressive given that BFKL runs on a single CPU. Furthermore, the DHRG improvement improves the BFKL embedding from dMAP 0.050 to dMAP 0.411, while the Lorentz embedding is improved from dMAP 0.192 to dMAP 0.320. This suggests that the layered approach of BFKL produces a better structure of the embedding. Furthermore, the combination of BFKL+DHRG is still more than 10 times faster than Lorentz 2D. (The dMAP result of 0.050 is very low compared to the continuous result of 0.284; this seems to be an outlier, in our other experiments the results of MAP and dMAP are very similar.)

Figure 2 shows our results for various benchmark datasets used in PE/LE, BFKL, and other sources. These include hierarchies: the WordNet verb hierarchy (VERBF), ACM and MeSH taxonomies, Stanford CS PhD network (De Nooy et al., 2018), the *tetrapoda* subtree from the tree of life project (Maddison et al., 2007). For all hierarchies, $u \to v$ iff $v$ is a superset (ancestor) of $u$; this is a transitive relation. In case of VERBF, we had to add an extra root node, ince BFKL requires the network to be connected. We also include real-world networks: the social circles from Facebook, scientific collaboration networks AstroPH, HepPH, CondMat and GrQC (Leskovec & Krevl, 2014), disease relationships (Goh et al., 2007), protein interactions in yeast bacteria (Jeong et al., 2001), and the brain connectome data (Allard & Serrano, 2020). We have not included other networks used in BFKL benchmarks because they are too large for slower algorithms such as Poincaré and Lorentz embeddings. In LE, the Enron email corpus and the historical linguistics data are analyzed using weighted edges, so we cannot compare them to BFKL or DHRG. Detailed results, including MeanRank and greedy routing stretch scores, are included in Appendix B.

Surprisingly, while BFKL has been designed specifically for scale-free networks and greedy routing and Lorentz embeddings have been benchmarked on hierarchies and mAP and MeanRank, our results show that BFKL or DHRG achieves significantly better results on many hierarchies (BFKL: NOUN,VERBF,MESH; DHRG: mesh,tetrapoda), while Lorentz embeddings tend to achieve better results on networks, especially for greedy routing (higher success rate and lower stretch). Still, the quality of BFKL, BFKL+DHRG, and Lorentz 2D embeddings turns out to be similar for the scale-free networks in our experiments, according to MeanRank and mAP. One counterexample in the YEAST network, where BFKL achieves significantly better results than Lorentz on mAP (0.756

vs 0.532). In all cases, BFKL (and even BFKL+DHRG) is orders of magnitude faster, making the Lorentz embeddings not practical on larger graphs. We also include the results of our experiments on three-dimensional Poincaré embeddings; these are still useful for visualization purposes (Munzner, 1998), especially in VR (Hart et al., 2017; Pisani et al., 2019; Weeks, 2021).

Our main focus is on bridging the gap between the two communities working on hyperbolic embeddings, so we concentrated on comparing the works from the time when the gap has appeared (BFKL, DHRG, Poincaré and Lorentz embeddings). We have also evaluated the classic HypViewer (Munzner, 1998) on hierarchies (if the hierarchy is not a strict tree, the parent is picked randomly); in most cases, MeanRank and mAP are quite low, although HypViewer aims to put similar nodes close, while due to how the transitive graphs are constructed for hypernymy hierarchies, high MeanRank and mAP measures are achieved when similar categories (e.g., "lion" and "tiger") are closer to their hypernyms (feline, mammal, animal, entity) than to each other, which promotes longer edges on the outer levels of the hierarchy, and shorter in the center. We have also evaluated the Mercator algorithm (García-Pérez et al., 2019), which is the standard tool used in the network community. The fast mode of Mercator usually produces worse embeddings than BFKL, while full Mercator usually achieves results between BFKL and Lorentz 2D. Unfortunately, the full Mercator is slower than Lorentz 2D for larger graphs. Unfortunately, the implementation of the higher-dimensional variant of Mercator (Jankowski et al., 2023) is not currently available, so we could not compare it to Poincaré 3D. In TreeRep (Sonthalia & Gilbert, 2020), it is proposed that, instead of learning a hyperbolic embedding, we should instead learn a tree. We agree with this proposition for tree-like hierarchies, but for networks such as FACEBOOK and the connectomes, hyperbolic embeddings achieve significantly better results. (Hyperbolic plane is tree-like in large scale and Euclidean-like in small scale, and thus may potentially combine the advantages of both approaches). LTiling (Yu & De Sa, 2019) did not generally achieve better results than Lorentz 2D in our experiments, while being significantly slower (contrary to DHRG, tiles are used only to improve numerical precision, not to make the process faster); however, this might be due to incorrectly set hyperparameters or testing on smaller, more shallow hierarchies, so the numerical precision issues did not yet become relevant. The hMDA method from (Sala et al., 2018) looks interesting, but it depends on the scaling factor, and it is not clear how to learn this parameter. Product space embeddings (Gu et al., 2019) are an interesting approach, but they use higher-dimensional spaces, so they cannot be compared to 2D methods (achieving better results can be explained with higher dimensionality). We would like to mention the recent work (Anonymous, 2023) on embedding into (three-dimensional) Thurston geometries using tiles and simulated annealing; on connectomes, this method yields better $\mathbb{H}^2$ and $\mathbb{H}^3$ embeddings than all methods studied in this paper. Most embedders are randomized, so we have repeated a portion of experiments using different seeds; this does not usually change the rankings (Appendix E).

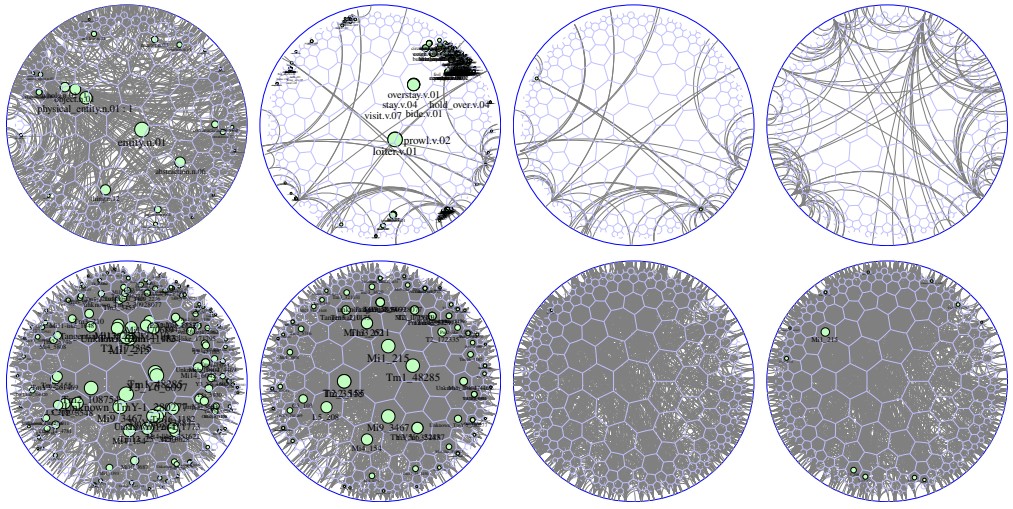

Figure 3: Top row: NOUN (Lorentz 2D). VERB (left to right: Lorentz 2D, Lorentz 2D+DHRG, BFKL). Bottom row: DROSOPHILIA1 (Lorentz 2D, Lorentz 2D+DHRG, BFKL, BFKL+DHRG).

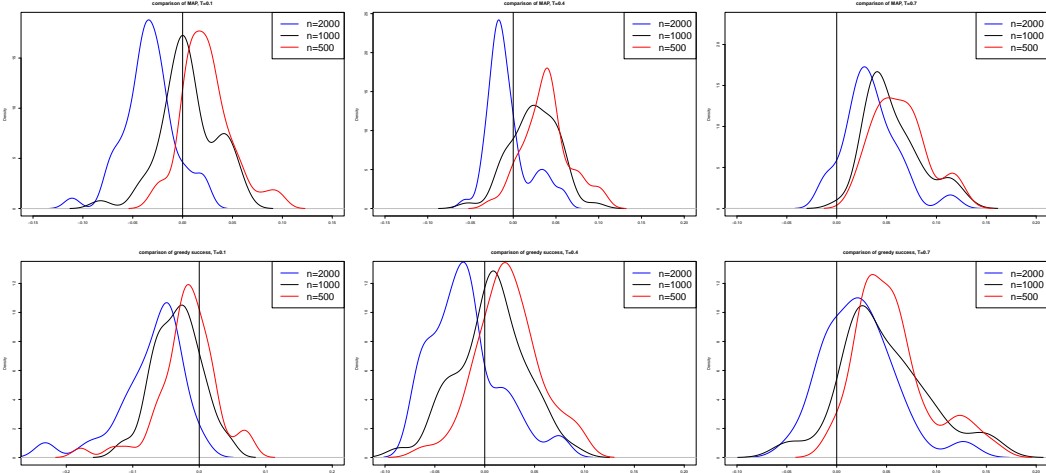

Figure 4: Density plots of the differences between the values of quality measures (mAP and greedy success) obtained by Lorentz 2D and BFKL embedders. Negative values indicate that BFKL embedder performed better.

## 4 VISUALIZATION

One application of 2D embeddings is visualization. We rendered the embeddings using the tools from DHRG; see Figure 3. All pictures are in Poincaré model, centered on the center of the hyperbolic disk used for embedding. One observation is that Lorentz embeddings tend to put nodes close to the center, while the center is generally avoided in BFKL, and DHRG improves the balance.

## 5 COMPARISON ON ARTIFICIAL SCALE-FREE NETWORKS

For a more statistical analysis, we have also compared BFKL and Lorentz 2D embeddings on artificially generated scale-free networks. We use the generator from BFKL based on the HRG model, with default $\alpha = 0.75$, network sizes $n \in \{500, 1000, 2000\}$ and temperature $T \in \{0.1, 0.4, 0.7\}$.

Fig 10 depicts the densities of the differences between the values of quality measures obtained by Lorentz 2D and BFKL embedders, and Table 1 contains results of the logit regressions on the determinants of the probability that BFKL embedder would perform better than Lorentz 2D embedder in terms of a given quality measure. No matter the quality measure, according to our results, the greater the graph, the higher probability that BFKL will perform better, however with rising temperature, that probability decreases. Real world network are considered to have fairly large values of $T$, such as $T = 0.7$ used for Internet mapping (Bläsius et al., 2016; Boguñá et al., 2010), which is consistent with our results on real-world scale-free networks. Although our models were aimed at intepretation instead of prediction, we included information of the prediction quality, both from cross-validation and benchmark. Both models are of a satisfactory quality.

Even if our results suggest that in many cases Lorentz 2D embedder outperforms BFKL embedders, it still comes at a high time cost. In Fig 5 we present trade-off between the markup in time expenditure (how many times longer it takes to compute) in comparison to BFKL and the percentage gain in the quality of the embedding (measured with MAP) resulting from using Lorentz 2D embedder. We conclude that there is no significant monotonic relationship between the time spent and the percentage gain in quality (p-values in Kendall-tau significance tests, as we encounter ties in our data that may make Spearman's rho inappropriate to use, are: 0.5282, 0.3141, and 0.0103 if we control for temperature 0.1, 0.4, and 0.7, respectively. The last result is insignificant at 1% level of significance).

|  | MAP | | greedy | |
|---|---|---|---|---|
|  | Coefficient | $\Pr(> |z|)$ | Coefficient | $\Pr(> |z|)$ |
| Intercept | -1.9583 | 9.11e-08 | 0.7312 | 0.00468 |
| Temp=0.4 | -0.8864 | 0.004922 | -2.0924 | 4.27e-12 |
| Temp=0.7 | -4.6115 | 1.59e-14 | -3.8869 | <2e-16 |
| Size = 1000 | 1.5956 | 0.000119 | 0.8173 | 0.00796 |
| Size = 2000 | 4.0095 | <2e-16 | 2.4526 | 6.34e-13 |
| N | 450 | | 450 | |
| $ACC_{cv}$ | 0.8598 | | 0.8008 | |
| $ACC_{bench}$ | 0.7178 | | 0.5289 | |
| $\kappa$ | 0.6288 | | 0.5992 | |

Table 1: Results of logit regressions for the determinants of BFKL embedder outperforming Lorentz 2D embedder in terms of quality measures. $ACC_{cv}$ and $\kappa$ are average accuracy and Kappa from 10-fold cross-validation; $ACC_{bench}$ is the accuracy of the naive model (always predict mode).

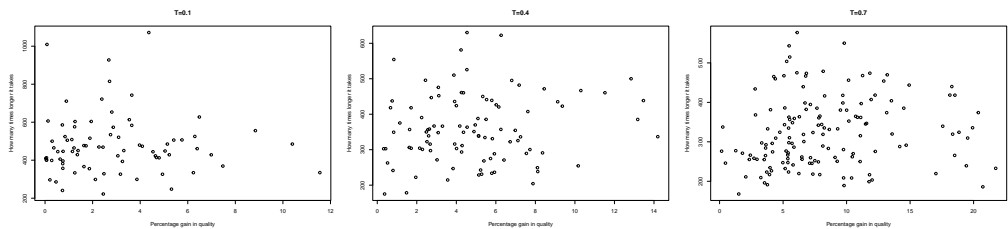

Figure 5: Comparisons of percentage gains in quality of the embedding against the markup in time expenditure in comparison to BFKL embedder.

## 6 CONCLUSION

We have compared the BFKL embedder against 2D Lorentz embeddings. Our main motivation for this comparison is the apparent lack of awareness of the algorithmic results on hyperbolic embeddings in the ML community. In all experiments, the BFKL embedder runs significantly (about 100 times) faster, while achieving results generally of similar quality, although in some cases one or the other embedder may get noticeably better results, depending on the input graph and the quality measure. Higher-dimensional Lorentz embedding generally gets better results than both kinds of 2D embeddings, even in three dimensions.

We have also found discrepancies between our results and the results in Nickel & Kiela (2017; 2018) that we are not unable to explain. In particular: in Nickel & Kiela (2017) 200-dimensional SGD Euclidean embeddings are performing worse than even low-dimensional Poincaré embeddings, but in our experiments, they consistently achieve significantly higher results; in Nickel & Kiela (2018) Lorentz embeddings achieve significantly better results than Poincaré, while in our experiments, where their performance is similar, and Poincaré is sometimes better. We could not reproduce the ACM and MESH taxonomies used in Nickel & Kiela (2018) (the number of edges and even nodes is not consistent with the numbers given – we are using our own data in this paper). See Appendix for details.

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

## A    IMPLEMENTATION USED

We have downloaded the embedders from the following repositories:

- Poincaré and Lorentz: `https://github.com/facebookresearch/poincare-embeddings` (last commit on Sep 16, 2021)
- BFKL: `https://bitbucket.org/HaiZhung/hyperbolic-embedder/overview` (last commit on Sep 8, 2016)
- DHRG: `https://github.com/zenorogue/hyperrogue/tree/master/rogueviz/dhrg` (last commit on April 1, 2023)
- TreeRep: `https://github.com/rsonthal/TreeRep` (last commit on Jun 23, 2023)
- LTiling: `https://github.com/ydtydr/HyperbolicTiling_Learning` (last commit Mar 19, 2020)
- HypViewer: `https://graphics.stanford.edu/~munzner/h3/download.html` (last modified in 2003)
- Mercator: `https://github.com/networkgeometry/mercator` (last commit Jun 21, 2022)

We have downloaded the connectome datasets from `https://github.com/networkgeometry/navigable_brain_maps_data`. The tree-of-life dataset has been included with DHRG.

## B    REAL-WORLD HIERARCHIES AND NETWORKS

The detailed results of our evaluation on real-world hierarchies can be found in table 2. We also include MAMMAL (the mammal subtree of Noun). The detailed results of our evaluation on real-world networks can be found in table 3. Figures 6, 7, 8, 9 contain visualizations of mAP, MeanRank, greedy success rate and greedy stretch ratio on those hierarchies and networks.

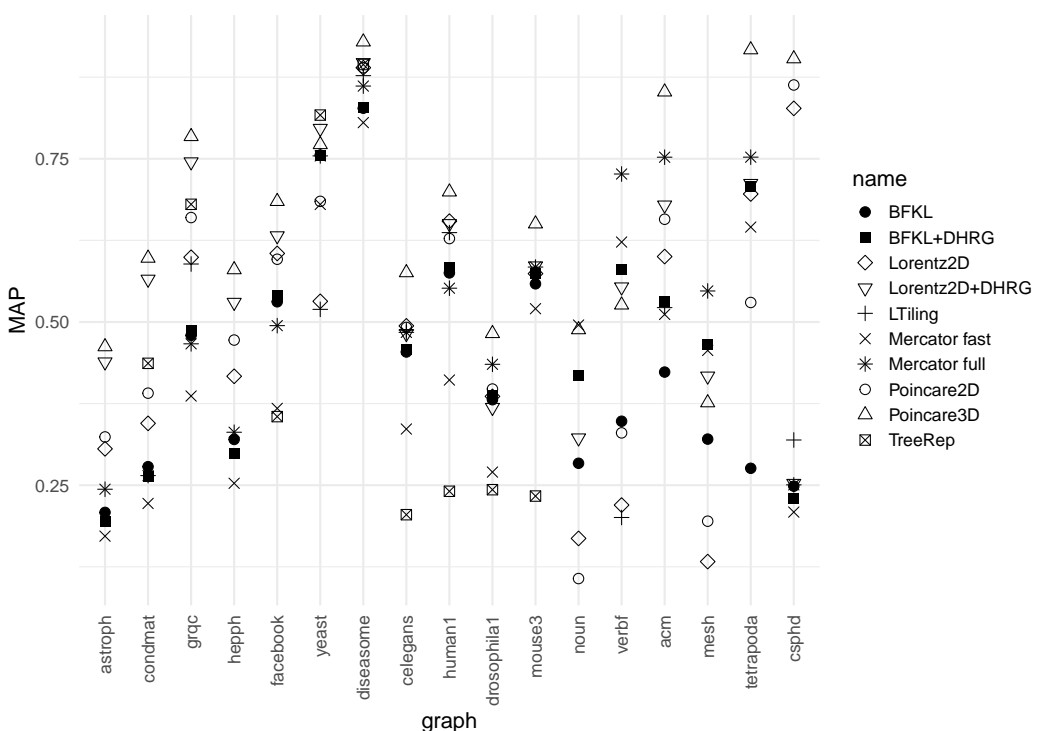

Figure 6: Quality assessment of embedders on real-world hierarchies and networks: mAP.

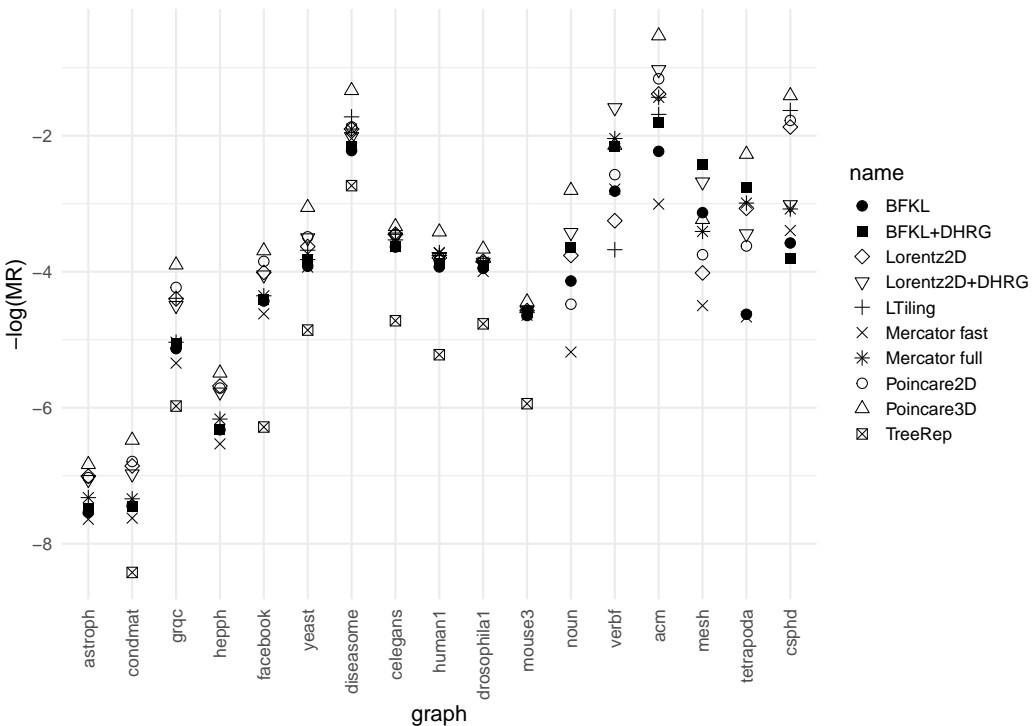

Figure 7: Quality assessment of embedders on real-world hierarchies and networks: -log(MeanRank).

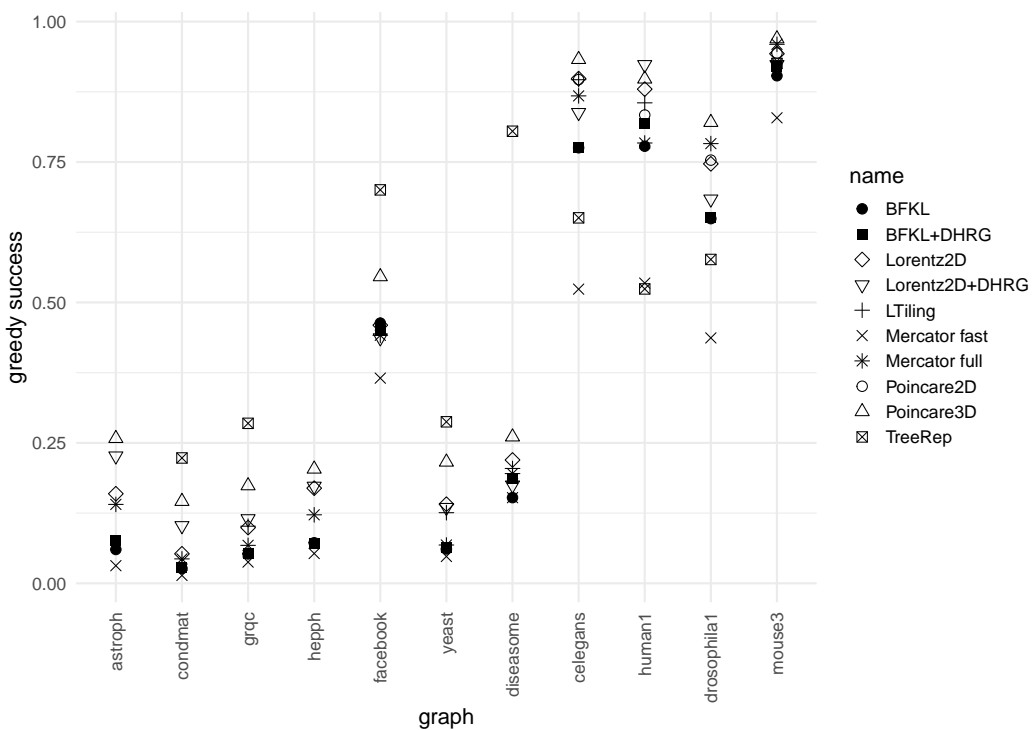

Figure 8: Quality assessment of embedders on real-world networks: greedy success rate.

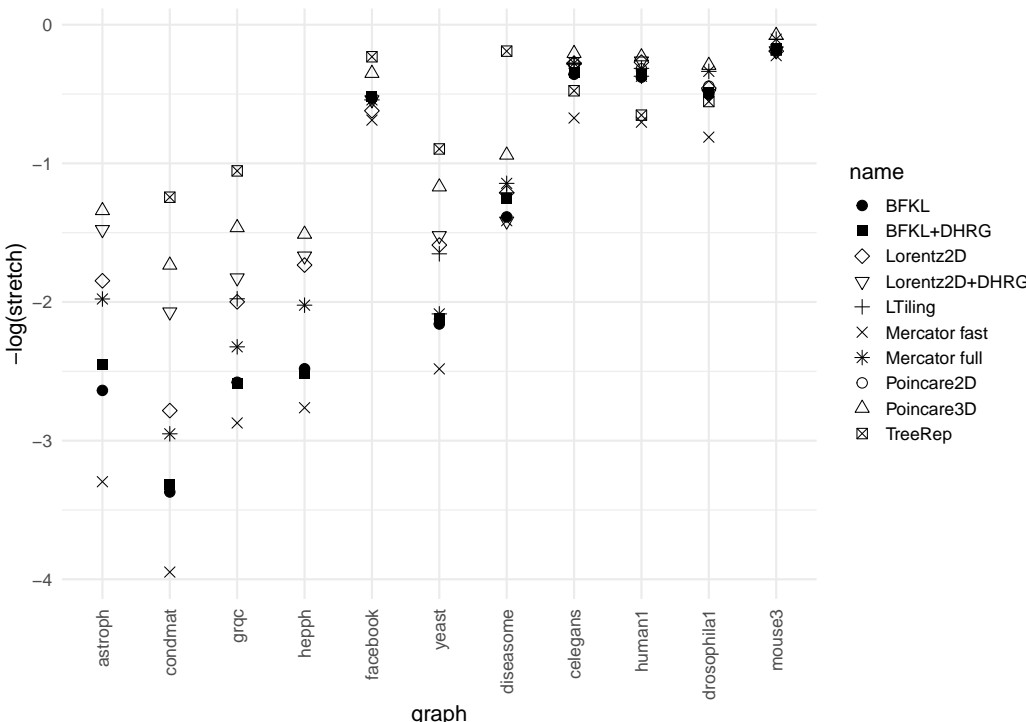

Figure 9: Quality assessment of embedders on real-world networks: -log(greedy stretch ratio).

| graph name | noun | mammal | verbf | acm | mesh | tetrap | csphd |
|---|---|---|---|---|---|---|---|
| nodes | 82115 | 1180 | 13543 | 2114 | 58737 | 11262 | 1025 |
| edges | 743086 | 6540 | 48621 | 8121 | 300290 | 527580 | 3978 |
| Lorentz 2D embed time [s] | 38578 | 269 | 2057 | 333 | 19832 | 19706 | 239 |
| Lorentz 2D eval time [s] | 1913.62 | 0.79 | 51.14 | 1.87 | 872.12 | 40.79 | 0.24 |
| BFKL embed time [s] | 428 | 66 | 37 | 4 | 342 | 738 | 2 |
| DHRG improve time [s] | 1209 | 4 | 563 | 5 | 849 | 858 | 12 |
| Poincare 2D embed time [s] | 0 | 0 | 2544 | 443 | 19137 | 26256 | 232 |
| Poincare 3D embed time [s] | 0 | 0 | 2778 | 457 | 20941 | 26648 | 233 |
| Mercator fast embed time [s] | 37202 | 18 | 914 | 38 | 16617 | 1770 | 2 |
| Mercator full embed time [s] | 0 | 41 | 4729 | 113 | 104459 | 4802 | 23 |
| ltiling embed time [s] | 0 | 0 | 63074 | 5613 | 0 | 0 | 2037 |
| Lorentz 2D radius AU | 14.509 | 14.509 | 14.509 | 13.290 | 14.509 | 14.509 | 14.509 |
| BFKL radius AU | 30.992 | 21.120 | 20.733 | 14.945 | 26.835 | 25.632 | 13.639 |
| HypViewer radius AU | 27.523 | 10.703 | 14.963 | 7.145 | 17.330 | 72.251 | 4.114 |
| RogueViz radius AU | 11.908 | 7.665 | 10.105 | 8.248 | 11.573 | 9.921 | 7.525 |
| Mercator fast radius AU | 53.492 | 19.895 | 38.833 | 25.654 | 49.917 | 38.554 | 26.238 |
| Mercator full radius AU | | 18.450 | 38.899 | 20.254 | 38.062 | 28.056 | 26.055 |
| ltiling radius | | | 13.285 | 12.508 | | | 13.885 |
| Lorentz 2D radius grid | 29 | 28 | 28 | 25 | 28 | 29 | 28 |
| BFKL radius grid | 44 | 40 | 43 | 29 | 44 | 42 | 27 |
| mAP Lorentz2D | 0.168 | 0.834 | 0.220 | 0.600 | 0.133 | 0.696 | 0.827 |
| dmAP Lorentz2D | 0.192 | 0.825 | 0.239 | 0.590 | 0.177 | 0.691 | 0.306 |
| dmAP Lorentz2D + DHRG | 0.320 | 0.850 | 0.543 | 0.675 | 0.414 | 0.715 | 0.256 |
| mAP Lorentz2D + DHRG | 0.322 | 0.855 | 0.554 | 0.679 | 0.417 | 0.712 | 0.253 |
| mAP BFKL | 0.284 | 0.219 | 0.348 | 0.423 | 0.321 | 0.276 | 0.248 |
| dmAP BFKL | 0.050 | 0.212 | 0.308 | 0.421 | 0.137 | 0.256 | 0.247 |
| dmAP BFKL + DHRG | 0.411 | 0.487 | 0.574 | 0.533 | 0.456 | 0.707 | 0.230 |
| mAP BFKL + DHRG | 0.418 | 0.488 | 0.580 | 0.532 | 0.466 | 0.707 | 0.229 |
| mAP landscape 200D | 0.302 | 0.437 | 0.465 | 0.480 | 0.346 | 0.586 | 0.215 |
| mAP landscape 50D | 0.055 | 0.258 | 0.176 | 0.281 | 0.084 | 0.262 | 0.183 |
| mAP Poincare 2D | 0.107 | 0.788 | 0.330 | 0.657 | 0.195 | 0.530 | 0.863 |
| mAP Poincare 3D | 0.488 | 0.950 | 0.526 | 0.852 | 0.376 | 0.917 | 0.903 |
| mAP Poincare 5D | 0.641 | 0.960 | 0.632 | 0.894 | 0.486 | 0.943 | 0.909 |
| mAP Euclidean50D | 0.921 | 0.999 | 0.923 | 0.999 | 0.824 | 0.997 | 1.000 |
| mAP Euclidean200D | 0.946 | 1.000 | 0.931 | 0.999 | 0.871 | 0.998 | 1.000 |
| mAP HypViewer | 0.047 | 0.124 | 0.134 | 0.134 | 0.122 | 0.014 | 0.416 |
| mAP RogueViz | 0.065 | 0.134 | 0.125 | 0.126 | 0.121 | 0.115 | |
| mAP Mercator fast | 0.495 | 0.695 | 0.622 | 0.512 | 0.456 | 0.645 | 0.209 |
| mAP Mercator full | | 0.841 | 0.727 | 0.752 | 0.548 | 0.752 | 0.251 |
| mAP ltiling | | | 0.201 | 0.522 | | | 0.319 |
| MR Lorentz2D | 43.0 | 1.8 | 25.8 | 4.0 | 55.6 | 21.5 | 6.5 |
| dMR Lorentz2D | 42.4 | 0.9 | 25.2 | 3.1 | 54.5 | 21.5 | 6.0 |
| MR Lorentz2D + DHRG | 30.7 | 1.9 | 4.9 | 2.8 | 14.6 | 31.3 | 20.4 |
| dMR Lorentz2D + DHRG | 28.9 | 1.0 | 4.1 | 1.8 | 13.9 | 29.1 | 17.7 |
| MR BFKL | 62.6 | 43.1 | 16.7 | 9.3 | 22.9 | 102.1 | 35.8 |
| dMR BFKL | 794.0 | 42.7 | 17.5 | 8.2 | 84.6 | 109.0 | 35.2 |
| dMR BFKL + DHRG | 38.2 | 8.8 | 7.6 | 5.0 | 9.9 | 15.2 | 42.5 |
| MR BFKL + DHRG | 38.1 | 10.0 | 8.6 | 6.1 | 11.3 | 15.9 | 45.0 |
| MR landscape 200D | 189.8 | 14.0 | 14.8 | 8.1 | 37.7 | 40.9 | 56.9 |
| MR landscape 50D | 1952.2 | 41.3 | 144.8 | 42.2 | 901.8 | 247.5 | 108.0 |
| MR Poincare 2D | 88.0 | 2.3 | 13.1 | 3.2 | 42.5 | 37.4 | 5.9 |
| MR Poincare 3D | 16.5 | 1.2 | 8.5 | 1.7 | 25.3 | 9.7 | 4.1 |
| MR Poincare 5D | 10.7 | 1.1 | 6.7 | 1.4 | 19.9 | 6.9 | 3.8 |
| MR Euclidean50D | 1.5 | 1.0 | 1.2 | 1.0 | 2.1 | 1.0 | 1.0 |
| MR Euclidean200D | 1.3 | 1.0 | 1.1 | 1.0 | 1.6 | 1.0 | 1.0 |
| MR HypViewer | 4452.4 | 145.7 | 276.1 | 77.0 | 522.2 | 5559.7 | 468.5 |
| MR RogueViz | 408.5 | 50.3 | 125.0 | 49.5 | 197.7 | 269.7 | |
| MR Mercator fast | 177.8 | 8.8 | 16.2 | 20.2 | | 106.1 | |
| MR Mercator full | | 2.8 | 7.7 | 4.2 | | 19.9 | |
| MR ltiling | | | 39.519 | 5.403 | | | 5.145 |

Table 2: Our results on real-world hierarchies.

| graph name | astrop | condma | grqc | hepph | facebo | yeast | diseas | celega | human1 | drosop | mouse3 |
|---|---|---|---|---|---|---|---|---|---|---|---|
| nodes | 17903 | 21363 | 4158 | 11204 | 4039 | 1458 | 516 | 279 | 493 | 350 | 1076 |
| edges | 393944 | 182572 | 26844 | 235238 | 176468 | 3896 | 2376 | 4574 | 15546 | 5774 | 181622 |
| Lorentz 2D embed time [s] | 14741 | 7349 | 986 | 8761 | 6317 | 183 | 130 | 204 | 639 | 255 | 6413 |
| Lorentz 2D eval time [s] | 88.98 | 120.31 | 4.98 | 37.18 | 5.37 | 1.05 | 0.28 | 0.15 | 0.29 | 0.19 | 0.81 |
| BFKL embed time [s] | 179 | 91 | 7 | 82 | 26 | 1 | 0 | 0 | 2 | 0 | 17 |
| DHRG improve time [s] | 323 | 436 | 64 | 74 | 15 | 25 | 5 | 8 | 1 | 1 | 27 |
| Poincare 2D embed time [s] | 0 | 0 | 0 | 0 | 0 | 235 | 155 | 254 | 0 | 0 | 0 |
| Poincare 3D embed time [s] | 0 | 0 | 0 | 0 | 0 | 235 | 156 | 260 | 0 | 0 | 0 |
| Mercator fast embed time [s] | 1565 | 2088 | 85 | 587 | 33 | 4 | 4 | 5 | 6 | 11 | 39 |
| Mercator full embed time [s] | 8906 | 15167 | 454 | 3797 | 408 | 47 | 7 | 7 | 9 | 13 | 65 |
| ltiling embed time [s] | 0 | 0 | 19721 | 0 | 0 | 3200 | 1592 | 2795 | 7307 | 0 | 7237 |
| Lorentz 2D radius AU | 11.651 | 10.941 | 10.962 | 12.335 | 11.555 | 9.563 | 11.142 | 6.483 | 11.664 | 7.713 | 10.493 |
| BFKL radius AU | 15.430 | 17.677 | 21.792 | 21.883 | 12.576 | 16.267 | 12.680 | 7.787 | 7.422 | 8.178 | 8.625 |
| Mercator fast radius AU | 70.081 | 64.709 | 42.147 | 53.510 | 31.315 | 22.056 | 26.306 | 16.438 | 20.775 | 23.148 | 28.747 |
| Mercator full radius AU | 56.490 | 52.331 | 40.502 | 51.965 | 30.507 | 25.973 | 24.869 | 16.261 | 15.998 | 24.585 | 30.338 |
| TreeRep diameter rec | | 21.902 | 21.095 | | 12.000 | 22.734 | 16.656 | 11.359 | 11.758 | 10.406 | 9.500 |
| TreeRep diameter norec | | 21.711 | 21.790 | | 13.406 | 26.836 | 17.000 | 9.000 | 12.344 | 12.000 | 8.000 |
| ltiling radius | | | 11.157 | | | 8.822 | 9.663 | 6.485 | 9.193 | | |
| Lorentz 2D radius grid | 22 | 21 | 20 | 23 | 22 | 18 | 21 | 12 | 21 | 14 | 20 |
| BFKL radius grid | 31 | 35 | 42 | 45 | 25 | 32 | 24 | 15 | 14 | 16 | 17 |
| mAP Lorentz2D | 0.306 | 0.345 | 0.599 | 0.417 | 0.605 | 0.532 | 0.889 | 0.494 | 0.654 | 0.386 | 0.574 |
| dmAP Lorentz2D | 0.307 | 0.353 | 0.595 | 0.447 | 0.602 | 0.522 | 0.881 | 0.482 | 0.649 | 0.372 | 0.568 |
| dmAP Lorentz2D + DHRG | 0.444 | 0.572 | 0.751 | 0.534 | 0.636 | 0.806 | 0.900 | 0.492 | 0.652 | 0.379 | 0.592 |
| mAP Lorentz2D + DHRG | 0.439 | 0.565 | 0.745 | 0.530 | 0.632 | 0.796 | 0.897 | 0.482 | 0.651 | 0.369 | 0.586 |
| mAP BFKL | 0.208 | 0.278 | 0.480 | 0.320 | 0.531 | 0.756 | 0.827 | 0.454 | 0.575 | 0.381 | 0.558 |
| dmAP BFKL | 0.207 | 0.276 | 0.466 | 0.318 | 0.525 | 0.750 | 0.824 | 0.447 | 0.569 | 0.377 | 0.553 |
| dmAP BFKL + DHRG | 0.195 | 0.262 | 0.492 | 0.294 | 0.541 | 0.750 | 0.826 | 0.460 | 0.587 | 0.383 | 0.576 |
| mAP BFKL + DHRG | 0.195 | 0.264 | 0.487 | 0.298 | 0.541 | 0.755 | 0.829 | 0.458 | 0.583 | 0.387 | 0.575 |
| mAP landscape 200D | 0.191 | 0.248 | 0.462 | 0.252 | 0.510 | 0.694 | 0.786 | 0.443 | 0.576 | 0.376 | 0.558 |
| mAP landscape 50D | 0.175 | 0.196 | 0.370 | 0.165 | 0.390 | 0.509 | 0.647 | 0.385 | 0.529 | 0.337 | 0.515 |
| mAP Poincare 2D | 0.324 | 0.391 | 0.660 | 0.472 | 0.596 | 0.685 | 0.889 | 0.492 | 0.628 | 0.397 | 0.576 |
| mAP Poincare 3D | 0.462 | 0.598 | 0.784 | 0.580 | 0.685 | 0.772 | 0.929 | 0.576 | 0.699 | 0.482 | 0.650 |
| mAP Poincare 5D | 0.512 | 0.662 | 0.815 | 0.628 | 0.713 | 0.845 | 0.932 | 0.600 | 0.728 | 0.519 | 0.670 |
| mAP Euclidean50D | 0.988 | 0.968 | 1.000 | 0.980 | 1.000 | 1.000 | 1.000 | 1.000 | 1.000 | 1.000 | 0.943 |
| mAP Euclidean200D | 0.994 | 0.975 | 1.000 | 0.984 | 1.000 | 1.000 | 1.000 | 1.000 | 1.000 | 1.000 | 1.000 |
| mAP Mercator fast | 0.172 | 0.222 | 0.387 | 0.253 | 0.368 | 0.680 | 0.805 | 0.336 | 0.411 | 0.270 | 0.520 |
| mAP Mercator full | 0.244 | 0.265 | 0.467 | 0.331 | 0.494 | 0.754 | 0.861 | 0.484 | 0.552 | 0.435 | 0.584 |
| mAP TreeRep rec orig | | 0.443 | 0.690 | | 0.398 | 0.820 | 0.902 | 0.228 | 0.269 | 0.263 | 0.284 |
| mAP TreeRep norec orig | 0.347 | 0.499 | 0.676 | | 0.407 | 0.820 | 0.864 | 0.223 | 0.278 | 0.277 | 0.290 |
| mAP TreeRep rec | | 0.437 | 0.680 | | 0.355 | 0.817 | 0.894 | 0.205 | 0.241 | 0.243 | 0.233 |
| mAP TreeRep norec | | 0.492 | 0.668 | | 0.360 | 0.816 | 0.852 | 0.204 | 0.259 | 0.250 | 0.227 |
| mAP ltiling | | | 0.589 | | | 0.519 | 0.877 | 0.488 | 0.637 | | |
| MR Lorentz2D | 1104.8 | 949.2 | 81.4 | 293.2 | 55.2 | 37.7 | 6.7 | 31.5 | 44.5 | 46.9 | 96.8 |
| dMR Lorentz2D | 1121.6 | 965.7 | 82.0 | 297.5 | 54.3 | 37.4 | 5.8 | 31.3 | 43.9 | 47.2 | 98.4 |
| MR Lorentz2D + DHRG | 1160.8 | 1069.2 | 89.8 | 320.8 | 57.3 | 33.2 | 7.5 | 32.8 | 44.3 | 46.5 | 96.5 |
| dMR Lorentz2D + DHRG | 1131.7 | 1043.9 | 86.1 | 310.8 | 53.7 | 31.2 | 6.4 | 30.8 | 42.6 | 45.5 | 93.5 |
| MR BFKL | 1880.0 | 1717.0 | 169.2 | 558.2 | 84.0 | 50.4 | 9.2 | 38.0 | 50.9 | 52.0 | 104.0 |
| dMR BFKL | 1919.3 | 1764.8 | 195.8 | 632.6 | 84.9 | 50.5 | 8.2 | 38.2 | 50.8 | 52.0 | 106.2 |
| dMR BFKL + DHRG | 1749.3 | 1715.6 | 155.9 | 558.7 | 80.0 | 44.9 | 7.6 | 36.4 | 46.7 | 48.3 | 98.0 |
| MR BFKL + DHRG | 1763.9 | 1719.2 | 156.5 | 558.6 | 82.1 | 45.9 | 8.7 | 37.6 | 48.6 | 49.7 | 99.7 |
| MR landscape 200D | 2020.4 | 2439.2 | 234.3 | 808.5 | 103.0 | 71.6 | 12.1 | 43.0 | 52.2 | 53.0 | 112.4 |
| MR landscape 50D | 2644.5 | 3748.7 | 396.2 | 1328.6 | 232.9 | 126.0 | 25.0 | 56.1 | 66.7 | 63.1 | 134.0 |
| MR Poincare 2D | 1127.0 | 889.4 | 68.8 | 302.2 | 46.9 | 32.7 | 6.5 | 31.6 | 46.4 | 47.2 | 96.5 |
| MR Poincare 3D | 929.5 | 650.0 | 49.4 | 242.1 | 40.1 | 21.2 | 3.8 | 28.1 | 30.4 | 39.1 | 85.0 |
| MR Poincare 5D | 725.9 | 492.5 | 36.5 | 206.4 | 32.6 | 13.7 | 3.4 | 25.0 | 24.7 | 35.1 | 80.1 |
| MR Euclidean50D | 1.0 | 1.0 | 1.0 | 1.0 | 1.0 | 1.0 | 1.0 | 1.0 | 1.0 | 1.0 | 15.3 |
| MR Euclidean200D | 1.0 | 1.0 | 1.0 | 1.0 | 1.0 | 1.0 | 1.0 | 1.0 | 1.0 | 1.0 | 1.0 |
| MR Mercator fast | 2081.3 | 2046.1 | 209.5 | 686.9 | 101.4 | 51.0 | 7.7 | 37.7 | 41.5 | 54.4 | 103.5 |
| MR Mercator full | 1513.3 | 1539.9 | 153.9 | 476.8 | 77.8 | 45.7 | 6.7 | 34.2 | 41.2 | 47.5 | 99.7 |
| MR TreeRep rec | | 4547.380 | 393.765 | | 535.833 | 128.848 | 15.367 | 112.256 | 185.004 | 117.506 | 380.246 |
| MR TreeRep norec | | 3986.390 | 423.601 | | 597.688 | 110.969 | 30.934 | 107.666 | 167.198 | 122.196 | 414.914 |
| MR ltiling | | | 80.730 | | | 39.755 | 5.563 | 31.524 | 43.010 | | |
| success BFKL | 0.060 | 0.026 | 0.052 | 0.072 | 0.463 | 0.061 | 0.153 | 0.775 | 0.778 | 0.649 | 0.904 |
| success BFKL + DHRG | 0.075 | 0.027 | 0.044 | 0.068 | 0.454 | 0.054 | 0.158 | 0.753 | 0.752 | 0.629 | 0.917 |
| success BFKL + DDHRG | 0.077 | 0.029 | 0.053 | 0.070 | 0.451 | 0.064 | 0.186 | 0.776 | 0.818 | 0.652 | 0.920 |
| success Lorentz2D | 0.159 | 0.053 | 0.099 | 0.169 | 0.460 | 0.141 | 0.220 | 0.899 | 0.880 | 0.747 | 0.943 |
| success Lorentz2D + DD | 0.227 | 0.103 | 0.115 | 0.173 | 0.437 | 0.135 | 0.175 | 0.838 | 0.923 | 0.684 | 0.923 |
| success Poincare2D | | | | | | | | 0.897 | 0.834 | 0.753 | 0.944 |
| success Poincare3D | 0.258 | 0.146 | 0.174 | 0.203 | 0.546 | 0.216 | 0.261 | 0.933 | 0.898 | 0.821 | 0.969 |
| success Mercator fast | 0.031 | 0.014 | 0.038 | 0.053 | 0.365 | 0.048 | 0.153 | 0.524 | 0.534 | 0.437 | 0.829 |
| success Mercator full | 0.140 | 0.043 | 0.068 | 0.122 | 0.442 | 0.068 | 0.195 | 0.868 | 0.784 | 0.783 | 0.960 |
| success TreeRep rec | | 0.223 | 0.285 | | 0.700 | 0.288 | 0.805 | 0.651 | 0.524 | 0.577 | 0.925 |
| success TreeRep norec | | 0.141 | 0.223 | | 0.828 | 0.306 | 0.777 | 0.613 | 0.541 | 0.684 | 0.877 |
| success ltiling | | | 0.102 | | | 0.126 | 0.205 | 0.897 | 0.855 | | |
| stretch BFKL | 13.98 | 29.09 | 13.18 | 11.96 | 1.70 | 8.66 | 4.00 | 1.43 | 1.46 | 1.65 | 1.20 |
| stretch BFKL + DHRG | 11.52 | 28.04 | 14.72 | 12.49 | 1.65 | 9.19 | 3.70 | 1.41 | 1.44 | 1.64 | 1.19 |
| stretch BFKL + DDHRG | 11.59 | 27.54 | 13.26 | 12.39 | 1.68 | 8.35 | 3.49 | 1.41 | 1.43 | 1.63 | 1.19 |
| stretch Lorentz2D | 6.34 | 16.17 | 7.38 | 5.66 | 1.86 | 4.90 | 3.36 | 1.32 | 1.31 | 1.58 | 1.21 |
| stretch Lorentz2D + DD | 4.39 | 7.94 | 6.21 | 5.31 | 1.72 | 4.58 | 4.12 | 1.36 | 1.30 | 1.65 | 1.21 |
| stretch Poincare2D | | | | | | | | 1.324 | 1.375 | 1.558 | 1.197 |
| stretch Poincare3D | 3.82 | 5.66 | 4.32 | 4.53 | 1.42 | 3.22 | 2.56 | 1.23 | 1.26 | 1.34 | 1.08 |
| stretch Mercator fast | 27.01 | 51.82 | 17.68 | 15.84 | 1.99 | 11.97 | 4.11 | 1.96 | 2.02 | 2.25 | 1.25 |
| stretch Mercator full | 7.23 | 19.12 | 10.21 | 7.56 | 1.72 | 8.05 | 3.14 | 1.32 | 1.45 | 1.40 | 1.11 |
| stretch TreeRep rec | | 3.469 | 2.866 | | 1.261 | 2.449 | 1.211 | 1.613 | 1.922 | 1.737 | 1.198 |
| stretch TreeRep norec | | 5.348 | 3.443 | | 1.172 | 2.565 | 1.195 | 1.647 | 1.871 | 1.553 | 1.233 |
| stretch ltiling | | | 7.223 | | | 5.222 | 3.362 | 1.333 | 1.373 | | |

Table 3: Our results on real-world networks.

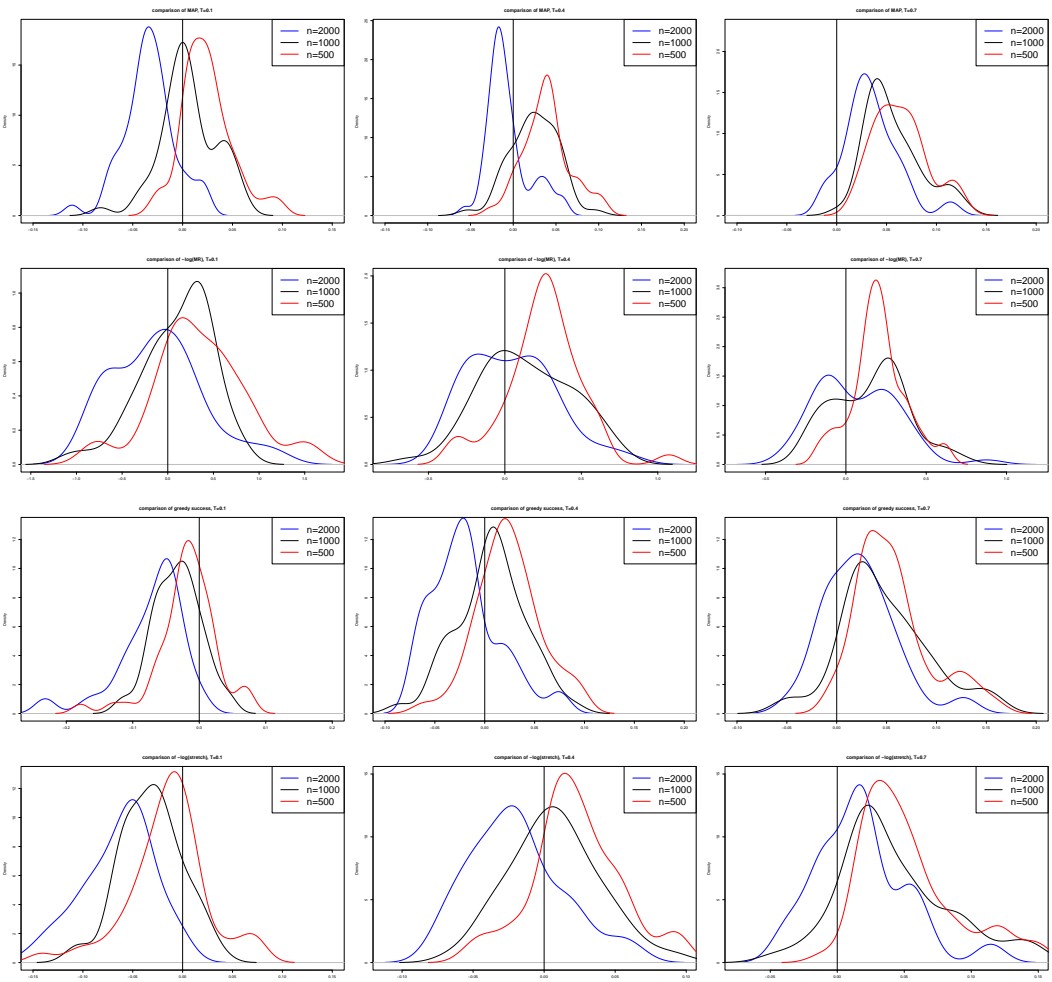

Figure 10: Density plots of the differences between the values of quality measures obtained by Lorentz 2D and BFKL embedders. Top to bottom: mAP, -log(MR), greedy success ratio, -log(greedy stretch ratio). Left to right: $T = 0.1$, $T = 0.4$, $T = 0.7$. Negative values indicate that BFKL embedder performed better.

## C  ARTIFICIAL NETWORKS

Table 4 and Figure 10 show the details of our evaluation of BFKL versus Lorentz 2D on artificial networks.

## D  DISCREPANCIES

In Table 5, our results are compared to the results obtained in Nickel & Kiela (2017; 2018). Note that VERB is different than VERBF used in our paper, which includes one extra node that is an ancestor of every other node.

Furthermore, the ACM hierarchy in Nickel & Kiela (2018) is given as 2299 nodes and 6526 edges, while ours has 2114 nodes and 8121 edges; and the MESH hierarchy is given as 28470 nodes and 191849 edges, while ours has 58737 nodes and 300290 edges.

| | MAP | | MeanRank | | Greedy success | | Greedy stretch | |
|---|---|---|---|---|---|---|---|---|
| | Coeff. | $\Pr(> \|z\|)$ | Coeff. | $\Pr(> \|z\|)$ | Coeff. | $\Pr(> \|z\|)$ | Coeff. | $\Pr(> \|z\|)$ |
| Intercept | -1.9583 | 9.11e-08 | -1.5132 | 2.11e-08 | 0.7312 | 0.00468 | 0.4256 | 0.09397 |
| Temp=0.4 | -0.8864 | 0.004922 | -0.3956 | 0.124942 | -2.0924 | 4.27e-12 | -2.1689 | 2.09e-12 |
| Temp=0.7 | -4.6115 | 1.59e-14 | -0.5732 | 0.028647 | -3.8869 | <2e-16 | -4.0383 | <2e-16 |
| Size = 1000 | 1.5956 | 0.000119 | 1.1338 | 0.000113 | 0.8173 | 0.00796 | 1.0527 | 0.00103 |
| Size = 2000 | 4.0095 | <2e-16 | 1.8908 | 5.62e-11 | 2.4526 | 6.34e-13 | 2.7747 | 1.29e-14 |
| N | 450 | | 450 | | 450 | | 450 | |
| $\text{ACC}_{cv}$ | 0.8598 | | 0.6707 | | 0.8008 | | 0.8047 | |
| $\text{ACC}_{bench}$ | 0.7178 | | 0.6711 | | 0.5289 | | 0.5556 | |
| $\kappa$ | 0.6288 | | 0.1324 | | 0.5992 | | 0.6053 | |

Table 4: Results of logit regressions for the determinants of BFKL embedder outperforming Lorentz 2D embedder in terms of quality measures. $\text{ACC}_{cv}$ and $\kappa$ are average accuracy and Kappa from 20-fold cross-validation; $\text{ACC}_{bench}$ is the accuracy of the naive model (always predict mode).

| graph name | NOUN | VERB | ASTRO | COND | GRQC | HEPPH |
|---|---|---|---|---|---|---|
| Poincaré 2D MR (ours) | 88.0 | 15.7 | 1127.0 | 889.4 | 68.8 | 302.2 |
| Poincaré 2D MR | 90.7 | 10.7 | — | — | — | — |
| Poincaré 2D mAP (ours) | 0.107 | 0.314 | 0.324 | 0.391 | 0.660 | 0.472 |
| Poincaré 2D mAP | 0.118 | 0.365 | — | — | — | — |
| Lorentz 2D MR (ours) | 43.0 | 42.1 | 1104.8 | 949.2 | 81.4 | 293.2 |
| Lorentz 2D MR | 22.8 | 3.64 | — | — | — | — |
| Lorentz 2D mAP (ours) | 0.168 | 0.184 | 0.306 | 0.345 | 0.599 | 0.417 |
| Lorentz 2D mAP | 0.305 | 0.579 | — | — | — | — |
| Euclidean 50D MR (ours) | 1.5 | 1.2 | 1.0 | 1.0 | 1.0 | 1.0 |
| Euclidean 50D MR | 1281.7 | — | — | — | — | — |
| Euclidean 50D mAP (ours) | 0.921 | 0.908 | 0.988 | 0.968 | 1.000 | 0.980 |
| Euclidean 50D mAP | 0.140 | — | 0.376 | 0.356 | 0.522 | 0.434 |

Table 5: Our results compared with the results from Nickel & Kiela (2017; 2018).

## E  REPEATED EXPERIMENTS

In Tables 6, 7, 8 and 9 we list the results of repeated experiments on the NOUN hierarchy and the GRQC, textscyeast, MOUSE3, HUMAN1, DROSOPHILA1 and CELEGANS networks. In most cases, the differences are minor and do not affect the rankings.

| graph name | noun | r1/nou | r2/nou | r3/nou | r4/nou |
|---|---|---|---|---|---|
| mAP Lorentz2D | 0.168 | 0.171 | 0.169 | 0.172 | 0.170 |
| dmAP Lorentz2D | 0.192 | 0.191 | 0.193 | 0.193 | 0.193 |
| dmAP Lorentz2D + DHRG | 0.320 | 0.324 | 0.326 | 0.321 | 0.327 |
| dmAP BFKL | 0.050 | 0.049 | 0.049 | 0.049 | 0.048 |
| dmAP BFKL + DHRG | 0.411 | 0.469 | 0.450 | 0.442 | 0.438 |
| mAP Poincare 2D | 0.107 | 0.104 | 0.105 | 0.105 | 0.105 |
| mAP Poincare 3D | 0.488 | 0.485 | 0.489 | 0.493 | 0.490 |
| MR Lorentz2D | 43.0 | 43.2 | 41.7 | 42.4 | 42.5 |
| dMR Lorentz2D | 42.4 | 42.7 | 41.1 | 41.9 | 42.0 |
| dMR Lorentz2D + DHRG | 28.9 | 28.6 | 27.2 | 29.4 | 28.0 |
| dMR BFKL | 794.0 | 772.5 | 809.9 | 839.3 | 821.7 |
| dMR BFKL + DHRG | 38.2 | 33.6 | 34.4 | 33.9 | 34.3 |
| MR Poincare 2D | 88.0 | 90.8 | 91.4 | 87.3 | 88.5 |
| MR Poincare 3D | 16.5 | 16.4 | 16.4 | 15.5 | 16.3 |

Table 6: Repeated experiments on the NOUN hierarchy.

| graph name | grqc | r1/grq | r2/grq | r3/grq | r4/grq | yeast | r1/yea | r2/yea | r3/yea | r4/yea |
|---|---|---|---|---|---|---|---|---|---|---|
| mAP Lorentz2D | 0.599 | 0.601 | 0.588 | 0.598 | 0.603 | 0.532 | 0.531 | 0.512 | 0.529 | 0.527 |
| dmAP Lorentz2D | 0.595 | 0.598 | 0.585 | 0.593 | 0.602 | 0.522 | 0.518 | 0.502 | 0.512 | 0.518 |
| dmAP Lorentz2D + DHRG | 0.751 | 0.753 | 0.745 | 0.752 | 0.754 | 0.806 | 0.798 | 0.768 | 0.811 | 0.783 |
| MR Lorentz2D | 81.4 | 71.2 | 77.6 | 74.4 | 71.9 | 37.7 | 39.0 | 37.9 | 39.2 | 38.0 |
| dMR Lorentz2D | 82.0 | 71.6 | 79.0 | 75.0 | 72.7 | 37.4 | 38.7 | 37.4 | 39.3 | 38.4 |
| dMR Lorentz2D + DHRG | 86.1 | 74.5 | 84.9 | 81.4 | 78.9 | 31.2 | 30.8 | 31.3 | 33.2 | 31.8 |
| dMR BFKL | 195.8 | 184.7 | 215.9 | 206.0 | 174.7 | 50.5 | 52.4 | 51.8 | 53.6 | 59.6 |
| dMR BFKL + DHRG | 155.9 | 146.2 | 169.7 | 160.4 | 131.5 | 44.9 | 45.7 | 45.4 | 48.2 | 56.3 |

Table 7: Repeated experiments on the GRQC and YEAST networks.

| graph name | mouse3 | r1/mou | r2/mou | r3/mou | r4/mou | human1 | r1/hum | r2/hum | r3/hum | r4/hum |
|---|---|---|---|---|---|---|---|---|---|---|
| mAP Lorentz2D | 0.574 | 0.572 | 0.574 | 0.575 | 0.570 | 0.654 | 0.633 | 0.644 | 0.643 | 0.651 |
| dmAP Lorentz2D | 0.568 | 0.567 | 0.568 | 0.568 | 0.563 | 0.649 | 0.627 | 0.637 | 0.637 | 0.646 |
| dmAP Lorentz2D + DHRG | 0.592 | 0.587 | 0.591 | 0.592 | 0.588 | 0.652 | 0.630 | 0.641 | 0.640 | 0.651 |
| dmAP BFKL | 0.553 | 0.559 | 0.558 | 0.557 | 0.553 | 0.569 | 0.536 | 0.567 | 0.582 | 0.557 |
| dmAP BFKL + DHRG | 0.576 | 0.581 | 0.579 | 0.579 | 0.575 | 0.587 | 0.555 | 0.580 | 0.599 | 0.572 |
| mAP Poincare 2D | 0.576 | 0.570 | 0.578 | 0.576 | 0.575 | 0.628 | 0.643 | 0.646 | 0.646 | 0.636 |
| mAP Poincare 3D | 0.650 | 0.654 | 0.652 | 0.650 | 0.652 | 0.699 | 0.722 | 0.718 | 0.715 | 0.715 |
| mAP Mercator fast | 0.520 | 0.520 | 0.520 | 0.520 | 0.520 | 0.411 | 0.411 | 0.411 | 0.411 | 0.412 |
| mAP Mercator full | 0.584 | 0.585 | 0.585 | 0.585 | 0.583 | 0.552 | 0.547 | 0.547 | 0.551 | 0.549 |
| mAP TreeRep rec orig | 0.284 | 0.271 | 0.294 | 0.275 | 0.299 | 0.269 | 0.298 | 0.284 | 0.290 | 0.302 |
| mAP TreeRep norec orig | 0.290 | 0.304 | 0.301 | 0.283 | 0.316 | 0.278 | 0.273 | 0.291 | 0.282 | 0.307 |
| mAP TreeRep rec | 0.233 | 0.238 | 0.243 | 0.222 | 0.242 | 0.241 | 0.283 | 0.262 | 0.255 | 0.280 |
| mAP TreeRep norec | 0.227 | 0.257 | 0.241 | 0.233 | 0.259 | 0.259 | 0.249 | 0.266 | 0.257 | 0.286 |
| MR Lorentz2D | 96.8 | 96.6 | 96.2 | 96.2 | 97.1 | 44.5 | 40.3 | 39.0 | 38.6 | 39.9 |
| dMR Lorentz2D | 98.4 | 97.8 | 97.7 | 97.8 | 99.1 | 43.9 | 40.2 | 39.3 | 38.6 | 39.5 |
| dMR Lorentz2D + DHRG | 93.5 | 93.5 | 93.0 | 93.4 | 94.4 | 42.6 | 38.0 | 36.9 | 36.6 | 37.9 |
| dMR BFKL | 106.2 | 103.0 | 103.7 | 103.5 | 104.6 | 50.8 | 51.4 | 51.6 | 47.9 | 49.4 |
| dMR BFKL + DHRG | 98.0 | 96.8 | 96.4 | 97.1 | 97.4 | 46.7 | 48.5 | 47.9 | 43.3 | 47.3 |
| MR Poincare 2D | 96.5 | 97.1 | 96.2 | 96.8 | 96.3 | 46.4 | 40.0 | 39.8 | 39.9 | 43.1 |
| MR Poincare 3D | 85.0 | 84.6 | 84.2 | 84.4 | 84.2 | 30.4 | 25.3 | 25.6 | 26.9 | 25.7 |
| MR Mercator fast | 103.5 | 103.5 | 103.5 | 103.5 | 103.5 | 41.5 | 41.5 | 41.5 | 41.6 | 41.6 |
| MR Mercator full | 99.7 | 99.5 | 99.4 | 99.4 | 99.5 | 41.2 | 41.2 | 41.1 | 41.1 | 41.3 |
| MR TreeRep rec | 380.246 | 381.121 | 408.380 | 418.490 | 405.135 | 185.004 | 131.951 | 160.604 | 153.917 | 129.617 |
| MR TreeRep norec | 414.914 | 381.244 | 403.176 | 381.503 | 392.632 | 167.198 | 164.443 | 136.862 | 153.998 | 140.229 |
| success Poincare2D | 0.944 | 0.931 | 0.948 | 0.945 | 0.943 | 0.834 | 0.839 | 0.889 | 0.869 | 0.883 |
| success Poincare3D | 0.969 | 0.971 | 0.969 | 0.965 | 0.971 | 0.898 | 0.917 | 0.915 | 0.926 | 0.909 |
| success Mercator fast | 0.829 | 0.832 | 0.830 | 0.829 | 0.833 | 0.534 | 0.531 | 0.535 | 0.533 | 0.536 |
| success Mercator full | 0.960 | 0.958 | 0.961 | 0.960 | 0.962 | 0.784 | 0.785 | 0.819 | 0.799 | 0.795 |
| success TreeRep rec | 0.925 | 0.879 | 0.845 | 0.867 | 0.821 | 0.524 | 0.555 | 0.550 | 0.532 | 0.629 |
| success TreeRep norec | 0.877 | 0.840 | 0.820 | 0.882 | 0.861 | 0.541 | 0.568 | 0.589 | 0.584 | 0.494 |
| stretch Poincare2D | 1.197 | 1.209 | 1.195 | 1.198 | 1.197 | 1.375 | 1.352 | 1.328 | 1.337 | 1.348 |
| stretch Poincare3D | 1.08 | 1.08 | 1.08 | 1.08 | 1.08 | 1.26 | 1.23 | 1.24 | 1.24 | 1.24 |
| stretch Mercator fast | 1.25 | 1.25 | 1.25 | 1.26 | 1.25 | 2.02 | 2.03 | 2.01 | 2.02 | 2.01 |
| stretch Mercator full | 1.11 | 1.11 | 1.11 | 1.11 | 1.11 | 1.45 | 1.45 | 1.40 | 1.43 | 1.44 |
| stretch TreeRep rec | 1.198 | 1.229 | 1.287 | 1.320 | 1.316 | 1.922 | 1.888 | 1.843 | 1.902 | 1.722 |
| stretch TreeRep norec | 1.233 | 1.299 | 1.305 | 1.213 | 1.245 | 1.871 | 1.824 | 1.811 | 1.731 | 2.005 |

Table 8: Repeated experiments on the MOUSE3 and HUMAN1 connectomes.

| graph name | drosop | r1/dro | r2/dro | r3/dro | r4/dro | celega | r1/cel | r2/cel | r3/cel | r4/cel |
|---|---|---|---|---|---|---|---|---|---|---|
| mAP Lorentz2D | 0.386 | 0.388 | 0.386 | 0.398 | 0.391 | 0.494 | 0.491 | 0.488 | 0.482 | 0.500 |
| dmAP Lorentz2D | 0.372 | 0.375 | 0.373 | 0.381 | 0.374 | 0.482 | 0.479 | 0.476 | 0.471 | 0.488 |
| dmAP Lorentz2D + DHRG | 0.379 | 0.391 | 0.385 | 0.401 | 0.401 | 0.492 | 0.490 | 0.486 | 0.485 | 0.500 |
| mAP Lorentz2D + DHRG | 0.369 | 0.381 | 0.376 | 0.392 | 0.396 | 0.482 | 0.480 | 0.476 | 0.475 | 0.490 |
| mAP BFKL | 0.381 | 0.388 | 0.389 | 0.386 | 0.376 | 0.454 | 0.469 | 0.454 | 0.462 | 0.471 |
| dmAP BFKL | 0.377 | 0.386 | 0.384 | 0.378 | 0.371 | 0.447 | 0.461 | 0.449 | 0.456 | 0.467 |
| dmAP BFKL + DHRG | 0.383 | 0.400 | 0.403 | 0.391 | 0.382 | 0.460 | 0.470 | 0.460 | 0.469 | 0.473 |
| mAP BFKL + DHRG | 0.387 | 0.397 | 0.401 | 0.392 | 0.380 | 0.458 | 0.467 | 0.460 | 0.465 | 0.469 |
| mAP Poincare 2D | 0.397 | 0.385 | 0.389 | 0.392 | 0.384 | 0.492 | 0.495 | 0.478 | 0.499 | 0.500 |
| mAP Poincare 3D | 0.482 | 0.483 | 0.488 | 0.488 | 0.473 | 0.576 | 0.575 | 0.571 | 0.575 | 0.583 |
| mAP Mercator fast | 0.270 | 0.271 | 0.270 | 0.270 | 0.270 | 0.336 | 0.337 | 0.337 | 0.337 | 0.336 |
| mAP Mercator full | 0.435 | 0.417 | 0.418 | 0.419 | 0.425 | 0.484 | 0.480 | 0.498 | 0.482 | 0.486 |
| mAP TreeRep rec orig | 0.263 | 0.241 | 0.245 | 0.260 | 0.274 | 0.228 | 0.218 | 0.206 | 0.241 | 0.236 |
| mAP TreeRep norec orig | 0.277 | 0.257 | 0.252 | 0.255 | 0.258 | 0.223 | 0.257 | 0.270 | 0.239 | 0.249 |
| mAP TreeRep rec | 0.243 | 0.222 | 0.219 | 0.236 | 0.245 | 0.205 | 0.190 | 0.188 | 0.213 | 0.198 |
| mAP TreeRep norec | 0.250 | 0.229 | 0.228 | 0.226 | 0.234 | 0.204 | 0.239 | 0.236 | 0.207 | 0.216 |
| MR Lorentz2D | 46.9 | 47.1 | 47.5 | 47.5 | 46.9 | 31.5 | 31.6 | 31.3 | 31.5 | 32.0 |
| dMR Lorentz2D | 47.2 | 47.2 | 47.3 | 47.5 | 47.0 | 31.3 | 31.6 | 31.2 | 31.5 | 31.8 |
| MR Lorentz2D + DHRG | 47.6 | 47.9 | 48.1 | 48.0 | 47.8 | 32.8 | 32.7 | 32.5 | 32.4 | 32.7 |
| dMR Lorentz2D + DHRG | 45.5 | 45.6 | 46.0 | 45.7 | 45.8 | 30.8 | 30.7 | 30.7 | 30.5 | 30.6 |
| MR BFKL | 52.0 | 54.4 | 53.5 | 52.4 | 52.9 | 38.0 | 36.4 | 39.5 | 36.9 | 36.8 |
| dMR BFKL | 51.9 | 53.8 | 53.1 | 52.4 | 52.4 | 38.2 | 36.4 | 39.5 | 36.5 | 36.7 |
| dMR BFKL + DHRG | 48.3 | 49.1 | 48.0 | 48.3 | 48.4 | 36.4 | 34.0 | 36.8 | 34.8 | 35.3 |
| MR BFKL + DHRG | 49.7 | 50.7 | 49.4 | 49.7 | 50.2 | 37.6 | 35.5 | 38.0 | 36.4 | 36.8 |
| MR Poincare 2D | 47.2 | 47.1 | 46.3 | 48.5 | 47.1 | 31.6 | 31.1 | 32.0 | 31.0 | 31.4 |
| MR Poincare 3D | 39.1 | 39.0 | 39.9 | 39.8 | 39.8 | 28.1 | 27.3 | 27.3 | 26.6 | 27.0 |
| MR Mercator fast | 54.4 | 54.3 | 54.3 | 54.3 | 54.3 | 37.7 | 37.8 | 37.8 | 37.8 | 37.8 |
| MR Mercator full | 47.5 | 47.8 | 48.0 | 47.9 | 47.7 | 34.2 | 34.4 | 34.0 | 34.1 | 34.3 |
| MR TreeRep rec | 117.506 | 123.417 | 134.651 | 129.946 | 122.314 | 112.256 | 117.753 | 104.263 | 113.346 | 111.039 |
| MR TreeRep norec | 122.196 | 125.698 | 111.734 | 126.610 | 120.817 | 107.666 | 103.975 | 100.043 | 91.454 | 104.623 |
| success BFKL | 0.649 | 0.641 | 0.623 | 0.618 | 0.614 | 0.775 | 0.796 | 0.763 | 0.774 | 0.796 |
| success BFKL + DHRG | 0.629 | 0.619 | 0.630 | 0.620 | 0.617 | 0.753 | 0.744 | 0.755 | 0.772 | 0.770 |
| success BFKL + DDHRG | 0.652 | 0.632 | 0.640 | 0.641 | 0.624 | 0.776 | 0.757 | 0.775 | 0.780 | 0.798 |
| success Lorentz2D | 0.747 | 0.727 | 0.742 | 0.758 | 0.742 | 0.899 | 0.891 | 0.889 | 0.871 | 0.894 |
| success Lorentz2D + DD | 0.684 | 0.662 | 0.658 | 0.688 | 0.696 | 0.838 | 0.834 | 0.843 | 0.831 | 0.849 |
| success Poincare2D | 0.753 | 0.715 | 0.757 | 0.723 | 0.743 | 0.897 | 0.896 | 0.874 | 0.903 | 0.898 |
| success Poincare3D | 0.821 | 0.814 | 0.826 | 0.844 | 0.820 | 0.933 | 0.943 | 0.925 | 0.933 | 0.958 |
| success Mercator fast | 0.437 | 0.439 | 0.448 | 0.433 | 0.445 | 0.524 | 0.525 | 0.525 | 0.522 | 0.525 |
| success Mercator full | 0.783 | 0.745 | 0.758 | 0.735 | 0.769 | 0.868 | 0.829 | 0.865 | 0.836 | 0.827 |
| success TreeRep rec | 0.577 | 0.580 | 0.596 | 0.695 | 0.708 | 0.651 | 0.603 | 0.560 | 0.670 | 0.571 |
| success TreeRep norec | 0.684 | 0.636 | 0.580 | 0.664 | 0.602 | 0.613 | 0.719 | 0.681 | 0.628 | 0.647 |
| stretch BFKL | 1.65 | 1.63 | 1.69 | 1.68 | 1.70 | 1.43 | 1.42 | 1.43 | 1.42 | 1.40 |
| stretch BFKL + DHRG | 1.64 | 1.63 | 1.62 | 1.65 | 1.67 | 1.41 | 1.42 | 1.41 | 1.40 | 1.40 |
| stretch BFKL + DDHRG | 1.63 | 1.63 | 1.65 | 1.65 | 1.69 | 1.41 | 1.43 | 1.40 | 1.40 | 1.40 |
| stretch Lorentz2D | 1.58 | 1.60 | 1.56 | 1.55 | 1.57 | 1.32 | 1.33 | 1.33 | 1.35 | 1.32 |
| stretch Lorentz2D + DD | 1.65 | 1.65 | 1.65 | 1.60 | 1.59 | 1.36 | 1.36 | 1.36 | 1.36 | 1.34 |
| stretch Poincare2D | 1.558 | 1.640 | 1.549 | 1.608 | 1.590 | 1.324 | 1.322 | 1.354 | 1.316 | 1.310 |
| stretch Poincare3D | 1.34 | 1.36 | 1.33 | 1.32 | 1.35 | 1.23 | 1.22 | 1.23 | 1.23 | 1.22 |
| stretch Mercator fast | 2.25 | 2.25 | 2.20 | 2.27 | 2.21 | 1.96 | 1.96 | 1.96 | 1.97 | 1.97 |
| stretch Mercator full | 1.40 | 1.45 | 1.43 | 1.45 | 1.41 | 1.32 | 1.36 | 1.32 | 1.37 | 1.37 |
| stretch TreeRep rec | 1.737 | 1.721 | 1.592 | 1.488 | 1.462 | 1.613 | 1.613 | 1.770 | 1.498 | 1.731 |
| stretch TreeRep norec | 1.553 | 1.580 | 1.753 | 1.547 | 1.652 | 1.647 | 1.433 | 1.483 | 1.668 | 1.538 |

Table 9: Repeated experiments on the DROSOPHILA1 and CELEGANS connectomes.

