# OpenReview forum: "Bridging ML and algorithms: comparison of hyperbolic embeddings"
_ICLR.cc/2024/Conference — Submitted to ICLR 2024_

### Official Review · Reviewer_dipB · 2023-10-27

**Soundness:** 2 fair
**Presentation:** 2 fair
**Contribution:** 2 fair
**Rating:** 3
**Confidence:** 3

**Summary:**

This paper aims to bridge the gap between two communities that have recently focused on hyperbolic embeddings: the algorithmic and the machine learning communities. It emphasizes that, surprisingly, there are few cross-references between them, even though the machine learning community could benefit from these algorithmic developments. The paper compares the timing of the contributions, the specificities (such as speed) of the algorithmic solutions, and the datasets on which the methods are benchmarked. It provides a comprehensive analysis in terms of formulations and presents an extensive experimental comparison, demonstrating that solutions provided by the algorithmic community are often faster while delivering similar performances.

**Strengths:**

The strengths of the paper are as follows:
- A focus on certain works that are mostly ignored by the machine learning community while relying on the same tools.
- A comprehensive analysis and comparison of the solutions, covering everything from formulation to experimental behavior.

In my opinion, the paper sheds light on somehow unknown works that exhibit interesting behavior in some cases. Consequently, the paper provides valuable insights into a topic that is of growing interest within the ML community.

**Weaknesses:**

The primary weakness of the paper is that the solutions are primarily benchmarked against 2D Lorentz and Poincaré embeddings. Although the content of the paper is intriguing, it is excessively specific, and I am not convinced of its overall significance. The paper fails to demonstrate how these results could enable the development of more efficient and faster solutions in general, especially in cases where higher-dimensional embeddings can be built. If there are some cases on which this should be done only on 2D, it should be emphasized.

The paper is sometimes difficult to follow, especially for readers like myself who come from the ML community. The explanation of the modeling of scale-free methods lacks clarity, and the experimental section is dense, making it difficult to grasp the big picture of the setup and the main conclusions.

**Questions:**

- What prevents benchmarking the solutions against higher dimensional Poincaré and Lorentz embeddings?

---

### Official Review · Reviewer_4BpB · 2023-10-27

**Soundness:** 3 good
**Presentation:** 3 good
**Contribution:** 3 good
**Rating:** 5
**Confidence:** 3

**Summary:**

This paper is about the lack of interaction between the ML community and TCS community. Specifically, that the ML community seems to be missing comparison to prior work on Hyperbolic Embedding and that these methods have some advantages over methods in the ML community.

**Strengths:**

The paper does a good job of thoroughly comparing Nickel and Kiela 2017 and 2018, with BFKL with and without the post processing step of DHRG. Specifically it shows the strengths of the TCS work.

The paper also does a very good job of listing related works and putting their work into context.

**Weaknesses:**

My main concern is that the paper does not present any new algorithmic methods, nor does it prove any theory results, nor does it obtain significant new insights into hyperbolic representation learning. It primarily serves as a benchmark paper that compares two classes of methods that have not been compared before.

I would suggest either submitting this to benchmarks track at NeurIPS or the new JMLR journal DMLR (Data-centric Machine Learning Research). I think this paper would be received very well in both of those venues.

**Questions:**

N/A

---

### Official Review · Reviewer_91h5 · 2023-10-31

**Soundness:** 2 fair
**Presentation:** 2 fair
**Contribution:** 2 fair
**Rating:** 3
**Confidence:** 2

**Summary:**

This paper reviews some classic hyperbolic embedding methods in the ML and algorithm community. The paper then compares these methods and claim that a classic (and earlier) method from the algorithm community achieves better performance than other methods, with a much faster computation speed.

**Strengths:**

This paper includes a very good review of the literature and investigated methods. It also makes effort to connect two different communities.

**Weaknesses:**

While this paper has some nice review and interesting comparison, it does not propose any novel idea and therefore the contribution is limited.

I cannot agree with the authors for their conclusion. Different algorithms are suitable for different tasks and data. While BFKL is faster, it is suitable for scale-free networks. The Poincaré and Lorentz embeddings, in contrast, are suitable for data with specific data structures, and their outputs can be further used for clustering or classification. They are also able to capture complex relationships in the data that go beyond the geometric structure of a scale-free network. In addition, using an embedding dimention $=2$ is unfair because Poincaré and Lorentz embeddings may have better performance at higher dimensions.

**Questions:**

None.

---

### Official Review · Reviewer_wq3r · 2023-10-31

**Soundness:** 3 good
**Presentation:** 3 good
**Contribution:** 3 good
**Rating:** 6
**Confidence:** 3

**Summary:**

The research work made the following contributions:

It compares hyperbolic embedding methods from the machine learning community (Poincaré and Lorentz embeddings) with methods from the algorithms community (BFKL embedding and DHRG improvement).

It finds that the BFKL embedding method is significantly faster (around 100x) than Poincaré and Lorentz embeddings while achieving competitive results on common evaluation metrics like Mean Rank and Mean Average Precision.

It shows that discretizing the embeddings using DHRG can further improve results for the BFKL method in some cases. DHRG also helps avoid numerical precision issues.


Overall, it demonstrates the potential for faster hyperbolic embedding methods from the algorithms literature to be competitive with popular machine learning approaches. More cross-pollination between communities could be beneficial.

**Strengths:**

This work highlights the potential for faster algorithmic hyperbolic embedding techniques to be competitive with popular ML methods.

**Weaknesses:**

The set of real-world networks tested is relatively small. Evaluating on a larger and more diverse set of networks could strengthen the empirical results.

The statistical analysis relates network factors like size and temperature to embedding quality, but the impact of other structural properties like clustering coefficient, degree distribution, etc., could also be enlightening.

The visualization analysis is limited. A more thorough qualitative analysis of the different embedding layouts and their strengths/weaknesses could provide additional insights.

**Questions:**

The discrepancy between your WordNet noun results and those reported in Nickel & Kiela 2017 is quite large. Could there be differences in the dataset/preprocessing used? Please investigate potential reasons for this discrepancy.

For the ACM and MeSH hierarchies, you note inconsistencies in the number of nodes/edges compared to Nickel & Kiela 2018. Can you clarify the source of the data used in your experiments? Is it possible you are using different versions of these hierarchies?

What is the largest network size you have managed to embed using the different methods? At what scale do you expect issues with the slower methods like Lorentz embeddings?

---

### Author Response · Authors · 2023-11-23

We thank all the reviewers for their insigthful comments, which will be helpful for us to improve the paper.

Indeed the purpose of our paper is not to introduce novel methods, although we believe it still has potential to be cited. We are very grateful to Reviewer 4BpB for the suggestions of other venues, we did not consider them previously.

We agree with Reviewer wq3r that the visualization analysis is limited, although we could not fit more due to the page limits.

Answering the questions of Reviewer wq3r: In the original BFKL paper (and the DHRG paper as well) networks of up to n=3764118 and m=16511741 are embedded, and according to the DHRG paper, the DHRG part takes 9335 seconds (which is much less than the time obtained by us for Lorentz embeddings for the noun hierarchy at n=82115 and m=743086). We have tried to find the reasons for the discrepancies (whether it could be a different version of the source data, etc.), but we could not explain them.

Answering the question of Reviewer dipB: we could benchmark against the higher-dimensional Poincaré and Lorentz embeddings but (from our current experience in other benchmark studies) the results would be trivial if using the quality heuristics from the literature -- the increase in the number of the dimensions usually comes with a higher than proportional increase in the quality of the embedding. This result stems from the artifact in optimization (reduced number of dimensions could be seen as imposing a restriction on that dimension, usually optimization without restrictions yields better that the restricted one) not the quality of the method itself (that we are primary interested in). To make benchmark "more fair" we would have to control properly for this artifact, e.g., by suggesting new information criteria that would use the information from the quality measure and add some penalty for the complexity of the embedding, e.g., the number of dimensions. However, so far we have not come up with a satisfactory idea for such information criteria.

---

### Meta-Review · Area_Chair_xEE3 · 2023-12-03

**Metareview:**

The authors made a literature review of hyperbolic embeddings in the machine learning community and its parallel development in the algorithm community. Based on the authors' experimental study, the latter (BFKL embedding, etc) is significantly faster with similar embedding metrics.

*Strength*:

The authors discovered a missing connection and some prehistory (that was not widely realized) between two versions of hyperbolic embeddings in the above communities. The paper is potentially interesting in developing new embeddings by combining the strengths of both communities.

*Weaknesses*:

The paper does not propose new embedding, or propose a new comparison protocol, or give enough theoretical insights, and therefore the contribution should be based on the empirical comparison.

The experiments conducted should be on a wider array of settings (not limited to 2D), more types of datasets, and a better presentation of the results, to make the comparison meaningful.

As highlighted by the reviewers, some parts (e.g. scale-free methods, experimental section) lack clarity and should be improved.

**Justification For Why Not Higher Score:**

As there is no new methods/theory proposed, the submission is assessed based on the empirical comparison. The overall quality of the experiment design and presentation of results should be further improved to support the claim of the authors.

**Justification For Why Not Lower Score:**

N/A

---

### Decision · Program_Chairs · 2024-01-16

Reject